



Atmospheric
Chemistry
and Physics

# Eddy flux measurements of sulfur dioxide deposition to the sea surface

**Jack G. Porter[1], Warren De Bruyn[2], and Eric S. Saltzman[1]**

[1]Department of Chemistry and Department of Earth System Science, University of California Irvine, Irvine, CA, USA
[2]Department of Chemistry and Biochemistry, Chapman University, Orange, CA, USA

**Correspondence:** Jack G. Porter (jgporter@uci.edu)

**Abstract.** Deposition to the sea surface is a major atmospheric loss pathway for many important trace gases, such as sulfur dioxide ($SO_2$). The air–sea transfer of $SO_2$ is controlled entirely on the atmospheric side of the air–sea interface due to high effective solubility and other physical–chemical properties. There have been few direct field measurements of such fluxes due to the challenges associated with making fast-response measurements of highly soluble trace gases at very low ambient levels. In this study, we report direct eddy covariance air–sea flux measurements of $SO_2$, sensible heat, water vapor, and momentum. The measurements were made over shallow coastal waters from the Scripps Pier, La Jolla, CA, using negative ion chemical ionization mass spectrometry as the $SO_2$ sensor. The observed transfer velocities for $SO_2$, sensible heat, water vapor, and momentum and their wind speed dependences indicate that $SO_2$ fluxes can be reliably measured using this approach. As expected, the transfer velocities for $SO_2$, sensible heat, and water vapor are lower than that for momentum, demonstrating the contribution of molecular diffusion to the overall air-side resistance to gas transfer. Furthermore, transfer velocities of $SO_2$ were lower than those of sensible heat and water vapor when observed simultaneously. This result is attributed to diffusive resistance in the interfacial layer of the air–sea interface.

## 1 Introduction

The deposition of soluble trace gases to the ocean surface is an important component in the global budgets of several important biogeochemical elements. For example, roughly 90–108 Tg yr$^{-1}$ of $SO_2$ is emitted to the atmosphere from fossil fuel combustion and industrial processes, from volcanic outgassing, and from the atmospheric photochemical oxidation of biogenic dimethylsulfide (DMS; Sheng et al., 2015; Chin et al., 2000). In the marine atmosphere, $SO_2$ oxidation contributes to the production and growth of aerosols which influence the Earth's radiation budget via aerosol backscatter of solar radiation and cloud optical properties. Global models estimate that dry deposition of $SO_2$ to the sea surface comprises slightly less than half of the total removal from the atmosphere (Sheng et al., 2015; Chin et al., 2000). The parameterization of dry deposition of soluble gases in atmospheric chemistry models is based largely on laboratory experiments, micrometeorological theory, or field studies in terrestrial environments (Liu et al., 1979; Liss, 1973; Mackay and Yeun, 1983). Relatively few direct flux studies of soluble trace gas deposition to the sea surface have been carried out due to a lack chemical sensors with sufficient sensitivity and response time for eddy covariance flux measurements. Faloona et al. (2009) reported air–sea eddy covariance surface fluxes for $SO_2$ using a fast-response chemical ionization mass spectrometric technique developed by Bandy et al. (2002). To our knowledge these are the only previous eddy covariance measurements of $SO_2$ surface fluxes over the ocean. Air–sea fluxes of the highly soluble organic compounds acetone and methanol have also been reported (Marandino et al., 2005; Yang et al., 2013, 2014, 2016).

In this study, we made eddy covariance flux measurements of $SO_2$ deposition to the coastal ocean from the Scripps Institute of Oceanography pier in La Jolla, California. These measurements were accompanied by simultaneous measurements of air–sea fluxes of momentum, water vapor, and sen-

sible heat. The goals of this study were (1) to directly determine the transfer coefficient of $SO_2$ and its wind speed dependence for comparison to existing estimates; (2) to compare the transfer coefficients of $SO_2$ with those of momentum, water vapor, and sensible heat to assess the relative importance of turbulent and diffusive resistance to $SO_2$ deposition; and (3) to attempt to detect the dependence of soluble gas deposition on molecular diffusivity in the marine environment.

## 2 Background

### 2.1 Air–sea gas transfer of highly soluble gases

Gas transfer across a gas–liquid interface is commonly parameterized as follows:

$$F = K \left( \frac{C_w}{\alpha} - C_a \right), \tag{1}$$

where $F$ is the air–sea flux (mol m$^{-2}$ s$^{-1}$), $C_a$ and $C_w$ are bulk air- and water-side concentrations (mol m$^{-3}$), and $\alpha$ is the dimensionless solubility ($C_w/C_a$ at equilibrium). $K$ represents the bulk gas transfer coefficient reflecting the physical processes limiting exchange on both sides of the interface, expressed in air-side units (m s$^{-1}$). The reciprocal of $K$, or resistance, can be partitioned into water-side and air-side processes, where

$$K^{-1} = R_{total} = r_w + r_a = \frac{1}{k_w} + \frac{\alpha}{k_a}. \tag{2}$$

In the case of gases like $SO_2$ with very high effective solubility ($\alpha \gg 1$) (Liss, 1971; Liss and Slater, 1974) and negligible seawater concentration (see below), the air side dominates the total resistance (i.e., $r_a \gg r_w$) so the gas transfer equation becomes

$$F = k_a \left( \frac{[SO_2]_w}{\alpha} - [SO_2]_a \right) \approx k_a [SO_2]_{air}, \tag{3}$$

where $k_a$ is the air-side gas exchange coefficient (m s$^{-1}$), also referred to as the deposition velocity. The transfer coefficient, $k_a$ (hereafter referred to as $k_{SO_2}$), encapsulates the physical processes controlling transport across the marine atmospheric surface layer to the air–sea interface. This transport is governed by (1) turbulence in the surface layer, (2) molecular diffusion close to the sea surface where turbulence is suppressed by molecular viscosity, and (3) the resistance to transfer across the air–sea interface at the water surface (Liss and Slater, 1974; Slinn et al., 1978). The transfer coefficient can be expressed in terms of resistance to deposition, as follows:

$$k_a^{-1} = r_{total} = r_{turbulence} + r_{diffusion} + r_{surface}. \tag{4}$$

The turbulent resistance term, sometimes referred to as aerodynamic resistance, is often approximated by the momentum transfer coefficient (or drag coefficient) under the assumption that there is no diffusive barrier to momentum transfer. Diffusive resistance is usually conceptualized in terms of the surface renewal model, involving periodic exchange of patches of near-surface air by turbulent eddies, with deposition of a trace gas to the sea surface via non-steady-state diffusion (Higbie, 1935; Danckwerts, 1951). This model implies a dependency on molecular diffusivity, as follows:

$$r_{diffusion} \propto Sc^n, \tag{5}$$

where $Sc$ is the Schmidt number defined as the kinematic viscosity of air ($\nu$) divided by the molecular diffusion coefficient ($D$) of the gas in air and $n$ is a constant. Early studies of soluble gas deposition to the ocean suggested a $Sc^{2/3}$ dependence based on boundary layer theory (Slinn et al., 1978). Current gas transfer models parameterize gas transfer as a surface renewal process with a $Sc^{1/2}$ dependence (Fairall et al., 2000; Donelan and Soloviev, 2016). Laboratory experiments using water-side-controlled gases show $n$ ranging from 0.50 to 0.66 for smooth and rough flow conditions (Jahne et al., 1987).

Interfacial surface resistance, i.e., resistance to air–sea gas transfer arising from physical–chemical interactions in a molecular scale layer at the surface, is included here for completeness. We are aware of no evidence that such processes are important at clean water surfaces for molecules such as $SO_2$ or $H_2O$ (see Sect. 2.2.3). The sea surface is often "contaminated" by the presence of organic compounds and particulates collectively referred to as the sea surface (or marine) microlayer. One could hypothesize that a hydrophobic surface film of sufficient coverage and thickness could introduce resistance to the transfer of small polar molecules such as $SO_2$ or $H_2O$, but such effects have not yet been demonstrated. It is well known that the microlayer can alter the surface tension of the sea surface, dampening the formation of capillary waves and indirectly altering the turbulent and diffusive resistance to transfer of momentum and gases (Frew et al., 1990; Bock and Frew, 1993; Pereira et al., 2016).

## 2.2 Physical chemical properties of $SO_2$ relevant to gas transfer

The interpretation of the $SO_2$ air–sea flux measurements in this study is based on the following premises: (1) deposition of $SO_2$ is controlled entirely on the air side of the air–sea interface and (2) surface ocean waters are always highly undersaturated in $SO_2$ with respect to the overlying atmosphere. In this section we discuss the basis for these assumptions.

### 2.2.1 Effective solubility of $SO_2$ and the kinetics of ionic equilibria

Sulfur dioxide is not a highly soluble gas, but it has a very large effective solubility in aqueous solution at elevated pH because of the dissociation of aqueous $SO_2$ into bisulfite and

sulfite ions ($HSO_3^-$; $SO_3^{2-}$). Collectively, dissolved $SO_2$ and its ionized forms are referred to as S(IV). The equilibria governing the aqueous speciation of $SO_2$ are listed below, with equilibrium constants given for seawater at 298 K (Millero et al., 1989).

$$SO_2 \rightleftharpoons SO_2(aq) \tag{R1}$$

$$SO_2(aq) + H_2O \rightleftharpoons HSO_3^- + H^+ \tag{R2}$$

$$HSO_3^- \rightleftharpoons SO_3^{2-} + H^+ \tag{R3}$$

$$H_{SO_2} = \frac{[SO_2(aq)]}{P_{SO_2}} = 1.24 \, M \, atm^{-1} \tag{6}$$

$$K_1 = \frac{[HSO_3^-][H^+]}{[SO_2(aq)]} = 2.6 \times 10^{-2} \, M \tag{7}$$

$$K_2 = \frac{[SO_3^{2-}][H^+]}{[HSO_3^-]} = 7.4 \times 10^{-7} \, M \tag{8}$$

Combining these equilibria yields an effective $SO_2$ solubility, as follows:

$$H_{eff} = H_{SO_2} \left[ 1 + \frac{K_1}{[H^+]} + \frac{K_1 K_2}{[H^+]^2} \right]. \tag{9}$$

$H_{SO_2}$ is the Henry's law solubility (M atm$^{-1}$), $K_1$ and $K_2$ are equilibrium constants in Reactions (R2) and (R3), $R$ is the gas constant (L atm K$^{-1}$ mol$^{-1}$), and $T$ is temperature (K). At the pH of seawater, $H_{eff}$ is $1 \times 10^{-7} \, M \, atm^{-1}$.

As noted by Liss (1971), the kinetics of S(IV) ionization in seawater are rapid, occurring on timescales much shorter than those for transport across the water-side interfacial layer. Based on rate constants for the forward and reverse reactions comprising the equilibria listed above, the characteristic time for equilibration of dissolved $SO_2$ with the ionic forms of S(IV) is roughly $4.5 \times 10^{-4}$ s (Schwartz and Freiberg, 1981), while the timescale for diffusive transport through the interfacial layer on the water side is on the order of seconds (Hoover and Berkshire, 1969). Consequently, $SO_2$ behaves as a highly soluble gas during the air–sea exchange process.

### 2.2.2 Placing a limit on the surface ocean concentration of S(IV)

To our knowledge, there are no published measurements of surface ocean S(IV). Here we place an upper limit on surface ocean S(IV) based on rough estimates for the sources of S(IV) to the ocean and the oxidation kinetics of S(IV) in seawater. The sources of S(IV) to the surface ocean include (1) release of hydrogen sulfide ($H_2S$) from marine sediments or deep waters, followed by oxidation to S(IV); (2) atmospheric deposition of $SO_2$; (3) production of $H_2S$ in surface waters from hydrolysis of photochemically produced carbonyl sulfide (OCS) followed by oxidation; and (4) production of $H_2S$ in surface waters from particulates and/or organisms. For the sediment source, we take the upper

limit of about $10^{-1}$ mol m$^{-2}$ yr$^{-1}$ from the global compilation of sulfate reduction rates by Bowles et al. (2014). For the atmospheric source, an atmospheric $SO_2$ mixing ratio of 1 nmol mol$^{-1}$ and a deposition velocity of 0.02 m s$^{-1}$ yields a source of $2.6 \times 10^{-2}$ mol m$^{-2}$ yr$^{-1}$. The other sources are many orders of magnitude smaller, based on surface ocean distributions and laboratory hydrolysis rates of OCS (Elliott et al., 1987; Cutter and Krahforst, 1988; Radford-Knoery and Cutter, 1994). Assuming that all of these sources are delivered to a shallow mixed layer of 10 m depth yields an upper limit on the S(IV) production rate ($P_{S(IV)}$) of about $10^{-2}$ mol m$^{-3}$ yr$^{-1}$. For the open ocean, the S(IV) production rate is likely much lower, because the sulfide from sedimentary sulfate reduction is not released directly into the surface ocean. The kinetics of oxidation of S(IV) in seawater was measured in the laboratory by Zhang and Millero (1991). They report the following rate expression:

$$\frac{[S(IV)]}{dt} = k_{oxidation}[S(IV)]^2, \tag{10}$$

where [S(IV)] is the seawater concentration of S(IV) (M) and $k_{oxidation}$ is the S(IV) oxidation rate constant (M$^{-1}$ s$^{-1}$) with a value of $1 \times 10^4$ M$^{-1}$ min$^{-1}$. The steady-state surface ocean S(IV) can be calculated as a balance between sources and oxidation, as follows:

$$P_{S(IV)} = k_{oxidation}[S(IV)]^2, \tag{11}$$

$$S(IV) = \sqrt{\frac{P_{S(IV)}}{k_{oxidation}}}, \tag{12}$$

yielding a steady-state S(IV) concentration of roughly $6 \times 10^{-8}$ M. Based on the effective solubility of $SO_2$ in seawater, this represents an equilibrium $SO_2$ gas-phase mixing ratio of only 2 fmol mol$^{-1}$. That is roughly 3 orders of magnitude lower than typical atmospheric $SO_2$ levels over the ocean (De Bruyn et al., 2006; Bandy et al., 1992; Chin et al., 2000). Therefore, one can justifiably assume that the sea surface is highly undersaturated in $SO_2$ with respect to the overlying atmosphere. It follows that the bulk air–sea concentration difference for $SO_2$ is essentially equal to the air-side concentration (Eq. 3).

### 2.2.3 Surface resistance to $SO_2$ deposition

In order for the molecular interface between water and air to play a significant role in air–sea gas transfer, the surface must introduce a resistance comparable to that across the turbulent and viscous layers above it. The surface can be modeled as a diffusive air-side layer with a thickness ($L$) equal to the mean free path of $SO_2$ in air, about 120 nm. The resistance across a flat planar surface layer can be estimated as

$$r_{surf} = \frac{L}{\gamma D} = \frac{1.2 \times 10^{-7}}{\gamma \times 1.3 \times 10^{-5}} \approx \frac{10^{-2}}{\gamma} \, s \, m^{-1}, \tag{13}$$

where $\gamma$ and $D$ are the accommodation coefficient and molecular diffusion coefficient of $SO_2$, respectively (Fuller et al., 1966). The timescales associated with turbulent and diffusive transport can be estimated using the COAREG (Coupled Ocean–Atmosphere Response Experiment Gas) gas transfer model (Fairall et al., 2000). For a height of 10 m and a wind speed of $10\,\mathrm{m\,s^{-1}}$ under neutral conditions, COAREG yields the following:

$$r_{\mathrm{turb}} + r_{\mathrm{diff}} \cong \boxed{\mathrm{TS4}}\,10^2\,\mathrm{s\,m^{-1}}. \qquad (14)$$

An accommodation coefficient of $10^{-4}$ would therefore be required in order for resistance at the surface to be comparable to that of the turbulent and diffusive atmosphere above. Laboratory studies of $SO_2$ uptake into clean water droplets suggest that the mass accommodation coefficient is about 0.1 (Worsnop et al., 1989). At this value, the surface resistance is only about 0.1 % of the overall resistance. Thus, surface resistance is not expected to play a significant role in air–sea gas transfer across clean water surfaces. The same is likely true for $H_2O$, which is believed to have an accommodation coefficient near unity, although there is considerable scatter in laboratory experiments (Morita et al., 2004). As noted earlier, the possibility of additional surface resistance for either $SO_2$ or $H_2O$ due to the presence of natural organic marine microlayers cannot be evaluated due to lack of information about their properties.

## 3 Methods

### 3.1 Study site and experimental setup

This study was conducted at Scripps Pier located in La Jolla, California, during April 2014. The local meteorology is characterized by a daily westerly sea breeze with occasional frontal systems that generally approach from the northwest. The pier structure extends 330 m from shore in the west–northwest direction and the water depth at the end of the pier is approximately 10 m. The end of the pier extends roughly 100 m past seaward of breaking waves. Meteorological sensors and air inlets were mounted at the end of a moveable 6 m boom mounted on the northwest corner of the pier. The boom was positioned to extend approximately into the prevailing winds. The sensing regions of the eddy covariance flux package and the air intake for $SO_2$ detection were located approximately 10 m above the sea surface. The sensor height was corrected for changes in tidal range during the experiment. Instrumentation for sulfur dioxide detection, data acquisition, clean air generator, and pumps were located in a trailer located at the end of the pier. Three-dimensional winds and fast-response temperature measurements were measured using a Campbell CSAT 3 sonic anemometer, with data collection at 50 Hz. Water vapor and air density were measured using an open-path infrared gas analyzer (IRGA; LI-COR model LI-7500) at 5 Hz. The instrument was calibrated using

a dew point generator (LI-COR model LI-610). Sea surface temperature was measured using a temperature probe array mounted on the pier with 9 probes vertically spaced by about 1 m. The sea surface temperature was taken to be the shallowest probe not exposed to air. Mean air temperatures were obtained from the NOAA meteorological station at the end of the pier.

For $SO_2$ detection, the air sampling inlet was similar to that used by Bell et al. (2013) to measure DMS. The air inlet was a $0.25''$ O.D. PFA tee fitting mounted just behind the sonic anemometer sensing region. Air was drawn into the inlet at a flow rate of $8500\,\mathrm{cc\,min^{-1}}$ and dried by passage through two counterflow Nafion membrane driers (Perma Pure LLC model PD-625-24PP) connected in series just after the inlet. The air passed from the driers through a $0.25''$ O.D., 13 m long PFA Teflon tube to a chemical ionization mass spectrometer located in the trailer. In the trailer, $1000\,\mathrm{cc\,min^{-1}}$ of the $8500\,\mathrm{cc\,min^{-1}}$ airflow was drawn through the ionization source of the mass spectrometer. A $200\,\mathrm{cc\,min^{-1}}$ stream of ozonized dry air (Pen Ray UV lamp) was added to the $1000\,\mathrm{cc\,min^{-1}}$ prior to entry into the ionization source. A continuous flow of isotopically labeled gas standard ($^{34}SO_2$ in $N_2$) was injected into the sampled air stream at the inlet tee. This gas standard was delivered to the inlet from an aluminum high-pressure cylinder located in the trailer, at a flow rate ranging from 1 to $10\,\mathrm{cc\,min^{-1}}$ from a $1/8''$ O.D. PFA tube.

All flow rates were controlled and logged using mass flow controllers interfaced to a PC. Air for the Nafion counterflow driers and ozone generator was supplied by a pure air generator and compressor (Aadco model 737-11), located in the trailer. Pumping for the air inlet and ionization source was provided by a carbon vane pump (Gast model 1023).

### 3.2 $SO_2$ detection by chemical ionization mass spectrometry

Atmospheric $SO_2$ was detected using a laboratory-built chemical ionization mass spectrometer (CIMS) in negative ion mode. This instrument was described previously for positive ion measurements of dimethylsulfide (Bell et al., 2013). The instrument was modified for this study by replacing a set of conical declustering lenses with a multi-lens ion funnel of the design developed by Kelly et al. (2010). This resulted in an order of magnitude improvement in ion transmission over the prior configuration of the instrument. In the CIMS instrument, ionization was carried out in a $0.25''$ inch glass-lined stainless steel flow tube containing a $^{63}Ni$ foil at 430 Torr and room temperature, with an airflow rate of $1000\,\mathrm{cc\,min^{-1}}$. Ions from the source enter the declustering region containing the ion funnel through a $250\,\mu\mathrm{m}$ diameter pinhole. The ion funnel is 127 mm long and consists of 100 concentric rings decreasing in diameter from 25.4 to 1.5 mm (Kelly et al., 2010). A DC gradient of $3\,\mathrm{V\,cm^{-1}}$ was applied to transmit ions axially and two phases of radio frequency

(RF; 2 MHz, 150 V p-p) were applied so that adjacent rings in the funnel were 180° out of phase. The ion funnel was operated at a pressure of 1 Torr. Ions exit the ion funnel via a 1 mm orifice into the first stage of a differentially pumped Extrel quadrupole mass filter (19 mm). Ions are detected using a dynode, ion multiplier, pulse amplifier/discriminator, and counting electronics (National Instruments model USB 6343). Ion counts were logged locally by the mass spectrometer control software and retransmitted as analog signals in real time with a fixed 2 s delay. The analog signals were logged by the multichannel data logger along with data from the meteorological sensors. Sulfur dioxide was detected in negative ion mode as $SO_5^-$ ($m/z$ 112), which was generated using the following reaction scheme previously described by Thornton et al. (2002).

$$O_2^- + O_3 \rightarrow O_3^- + O_2 \tag{R4}$$

$$O_3^- + CO_2 \rightarrow CO_3^- + O_2 \tag{R5}$$

$$CO_3^- + SO_2 \rightarrow SO_3^- + CO_2 \tag{R6}$$

$$SO_3^- + O_2 + N_2 \rightarrow SO_5^- + N_2 \tag{R7}$$

The addition of ozone minimizes the competing reaction $O_2^- + SO_2 \rightarrow SO_4^-$ and increases response to $SO_2$ (Möhler et al., 1992). When operating the ionization source at atmospheric pressure there was interference at $m/z$ 112 from the $CO_4(H_2O)_2^-$ cluster ion. This was essentially eliminated by dropping the pressure in the source to 430 Torr.

Isotopically labeled $^{34}SO_2$ delivered to the air inlet served as an internal standard to account for any wall losses or variations in instrument sensitivity due to changes in ambient conditions. The flow rate of the gas standard was adjusted to achieve a $^{34}SO_2$ level of roughly 100 pmol mol$^{-1}$ after dilution into the ambient airflow. The gas standard was prepared in our laboratory in a high-pressure aluminum gas cylinder (Scott Marrin model 30A) and delivered via mass flow controller. These gas standards were calibrated in the lab against a gravimetrically calibrated permeation device using an inert dilution system described by Gallagher et al. (1997). The isotopically labeled standard was detected at $m/z$ 114. The ambient $SO_2$ mixing ratio was calculated from the field data as follows:

$$X_{SO_2} = \frac{S_{112}}{S_{114}} \frac{f_{std}}{f_{total}} X_{tank}, \tag{15}$$

where $S_{112}$ and $S_{114}$ are blank-corrected mass spectrometer signals, $f_{std}$ and $f_{total}$ are the gas flow rates of the isotopic standard and inlet, and $X_{tank}$ is the molar mixing ratio of $^{34}SO_2$ in the compressed cylinder. Because the air stream was dried in the inlet tube prior to analysis, $X_{SO_2}$ represents the mixing ratio of $SO_2$ in dry air. Blanks involved sampling air through a carbonate-impregnated filter to quantitatively remove ambient $SO_2$. Whatman 41 filters for this purpose were soaked in 1 % sodium carbonate solution and dried prior to use. During this study the $SO_2$ instrument exhibited sensitivity of approximately 150 Hz ppt$^{-1}$.

### 3.3 Flux data acquisition, post-processing, and gas transfer calculations

The analog data streams from the meteorological and chemical sensors were filtered with a Butterworth filter and logged at 50 Hz using a National Instruments multichannel data logger. Post-processing consisted of (1) aligning the data to account for instrumental electronic delays and the delay due to the airflow transit time through the inlet tube; (2) rotating the 3-D winds for each flux interval into the frame of reference of the mean winds and to account for tilt in the sonic anemometer (1.3°); (3) converting the data to geophysical units; (4) computing vertical fluxes of water vapor, sensible heat, $SO_2$ and momentum; (5) applying a high-frequency correction to the $SO_2$ fluxes to account for loss of fluctuations in the inlet tubing; and (6) applying various quality control criteria to filter the resulting data set for instrumental issues or unsuitable environmental conditions. Data processing was carried out using Matlab (Mathworks). The inlet delay for $SO_2$ was determined experimentally in the laboratory prior to field deployments to be roughly one second. The measured delay was consistent with the offset required for maximizing the covariance between vertical wind and $SO_2$ concentration. Sulfur dioxide was measured as a dry mixing ratio since the air stream was dried prior to entering the mass spectrometer and converted to concentration (mol m$^{-3}$) using the dry air density. Water vapor concentrations measured by the LI-COR IRGA were corrected to account for air density fluctuations and converted to concentration (mol m$^{-3}$). The saturation vapor pressure of seawater at the sea surface temperature was calculated following Sharqawy et al. (2010). The mean air temperature was corrected for the adiabatic lapse rate, and the sonic temperatures were corrected for humidity. $SO_2$, water vapor, temperature, and winds were corrected to 10 m height and neutral stability using COARE (Businger et al., 1971; Fairall et al., 1996; Edson et al., 2013; Fairall et al., 2003). The data set was subdivided into 13 min flux intervals for processing. The resulting data consisted of means and variances for air temperature, relative humidity, $SO_2$, and seawater surface temperature. Fluxes of momentum (Reynolds stress, $\tau$), water vapor, sensible heat, and $SO_2$ were calculated for each interval according to

$$F_{SO_2} = \overline{w'C'_{SO_2}}, \tag{16}$$

$$F_{H_2O} = \overline{\rho}\, \overline{w'X'_{H_2O}}, \tag{17}$$

$$F_{mom} = \overline{\rho}\, \sqrt{\overline{(w'u')^2} + \overline{(w'v')^2}}, \tag{18}$$

$$F_{SH} = \overline{\rho}\, c_p\, \overline{w'T'}, \tag{19}$$

where $u$, $v$, and $w$ are the winds; $c_p$ is the heat capacity of air and $\rho$ is air density in kg m$^{-3}$; and the other variables are defined previously. $T$ is the air temperature corrected for humidity and the adiabatic lapse rate. Primed quantities with overbars represent the ensemble average of the fluctuations about the mean.

Transfer velocities were computed following Eqs. (1) and (3), as follows TS5:

$$k_{SO_2} = -\frac{F_{SO_2}}{[SO_2]_a}, \tag{20}$$

$$k_{H_2O} = \frac{F_{H_2O}}{(\overline{X_s} - \overline{X_{H_2O}})\,\overline{\rho_{dry}}}, \tag{21}$$

$$k_{mom} = \frac{F_{mom}}{U_{10}\,\overline{\rho}}, \tag{22}$$

$$k_{SH} = \frac{F_{SH}}{(\overline{T_s} - \overline{T})\,\overline{\rho}\,\overline{c_p}}. \tag{23}$$

$X_s$ is the calculated mixing ratio of water vapor corresponding to the saturation vapor pressure of water at the sea surface temperature.

### 3.4 High-frequency correction for inlet tubing

High-frequency fluctuations in the mixing ratio of $SO_2$ are attenuated during the passage of ambient air through inlet tubing and membrane driers. The attenuation characteristics of the inlet used in this study were characterized by interrupting the addition of an $SO_2$ gas standard to the airflow, resulting in an exponential decay of the $SO_2$ signal. A decay constant ($K$) was obtained from the slope of a linear regression to a plot of $\log(SO_2)$ vs. time. The attenuation of the inlet was modeled as a first-order low-pass Butterworth filter with a cut-off frequency, $F_c = K/(2p)$, of about 1.5 Hz. A high-frequency correction factor or gain, $G$, was computed for each flux interval by applying the filter to the sonic temperature time series data and taking the ratio of the filtered and unfiltered fluxes as follows:

$$G = F_{unfiltered}/F_{filtered}. \tag{24}$$

Linear regression of the gain against wind speed yielded $G = 0.005U_{10} + 1.018$. The $SO_2$ flux for each interval was multiplied by the gain using this relationship and the mean wind speed for the interval.

### 3.5 Quality control criteria

Several quality control criteria were applied to the data to identify and eliminate flux intervals collected under unsuitable conditions or with instrumental problems. They are described as follows.

1. Co-spectral shape: a cumulative sum of co-spectral density, normalized to the total flux, was computed for each flux interval, summing from low to high frequency. Intervals were rejected if (a) the cumulative sum at 0.004 Hz exceeded the total flux or was opposite in sign or (b) the difference between the cumulative flux at two consecutive frequencies exceeded 18 %. These criteria identified most intervals with obvious deviations in co-spectral shape from those defined in Kaimal et al.

(1972). Most of these intervals were caused by electronic noise on the sonic anemometer signal.

2. Small air–sea differences: intervals with air–sea concentration differences close to the propagated uncertainty of the analytical measurements were eliminated. The criteria for water vapor, sensible heat, and $SO_2$ were $10^{-3}$ mol mol$^{-1}$, 0.7 °C, 10 pmol mol$^{-1}$.

3. Wind sector: intervals with mean wind directions deviating from onshore by more than ±90° were rejected.

4. Stable atmospheric conditions: intervals with stable atmospheric conditions, defined as $z/L > 0.07$, were rejected (Oncley et al., 1996).

5. Local $SO_2$ contamination: intervals with sharp excursions in $SO_2$ associated with local contamination due to nearby vessels were subjectively identified and rejected.

## 4 Observations

### 4.1 Meteorological and oceanic conditions

The field study was carried out from 6 to 27 April 2014. Time series of meteorological and oceanographic parameters and fluxes measured during this study are given in Fig. 1. Winds were generally light during the study, with a mean wind speed of $3.8 \pm 2.0$ m s$^{-1}$ and a range of 0–9.7 m s$^{-1}$. Air temperatures were $16.2 \pm 1.3$ °C with a range from 12.9 to 19.9 °C and the average relative humidity was 80 %. Sea surface temperatures averaged $16.5 \pm 0.9$ °C with a range of 13.8–18.3 °C. The $SO_2$ mixing ratio ranged from below detection to 560 pmol mol$^{-1}$ with a mean of $100 \pm 114$ pmol mol$^{-1}$. Sharp spikes in $SO_2$ were usually associated with military or commercial vessels passing upwind of the pier. Low $SO_2$ levels were associated with the occurrence of morning fog. For the first few days of the study, a high-pressure region was located over the study site (DOY 97–100), during which winds were light and air temperatures were warm. Air mass back trajectories from this period indicate that marine air masses flowed from the north, passing inland over California before reaching the site. $SO_2$ levels were relatively high during this time likely due to fossil fuel combustion. After the high-pressure system moved out of the region, airflow was from the northwest, arriving at the study site directly from the ocean, and $SO_2$ levels were relatively low during this period. There was a notable increase in wind speed starting at DOY 106. On DOY 115 a low-pressure system passed over the region with higher wind speeds.

The Scripps Pier site experiences a consistent diurnal sea breeze, with offshore flow during the evening and extending to the early morning. Data from periods with offshore flow were excluded from the analysis in the quality control process. Due to the sea breeze locally and along the coast, there

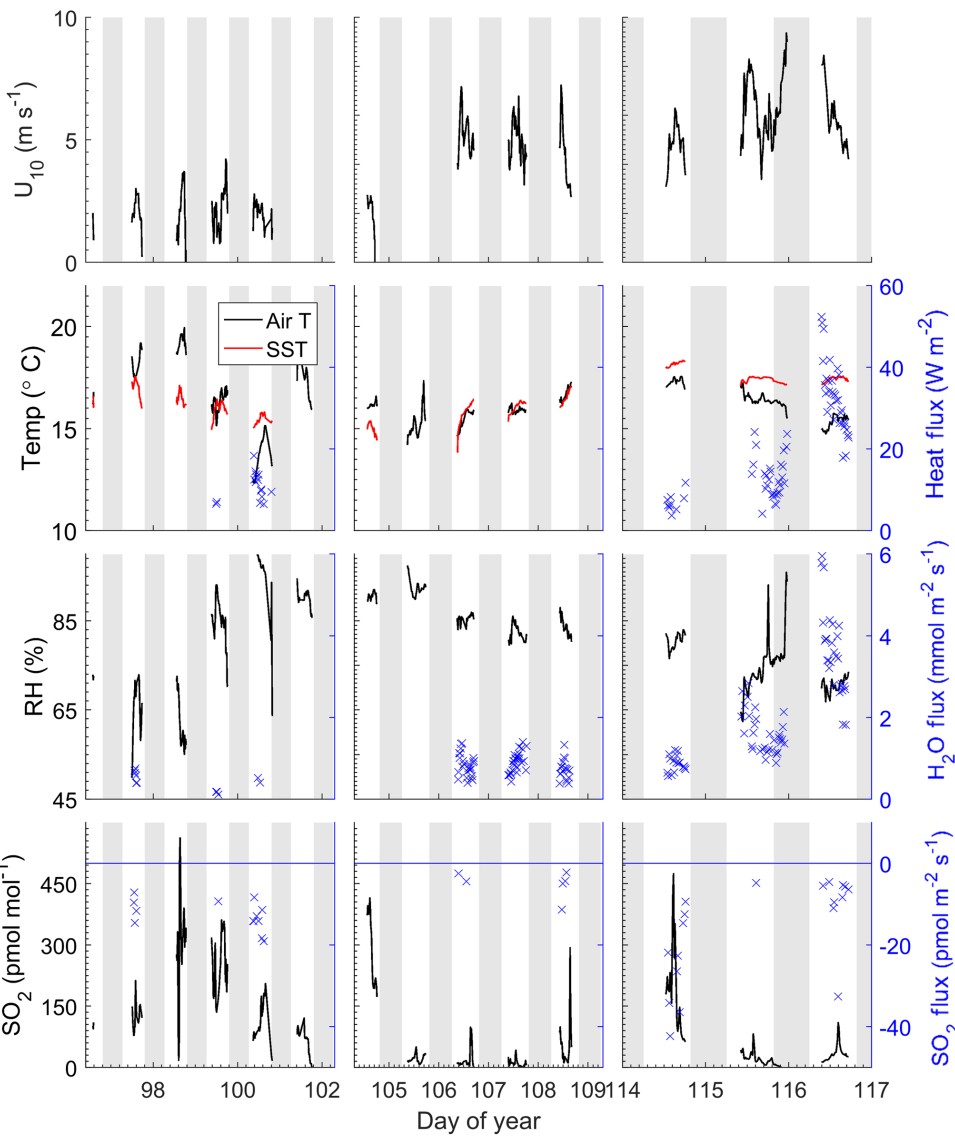

**Figure 1.** Time series of meteorological and oceanographic parameters measured on Scripps Pier during 6–27 April 2014. The grey bands indicate night. The blue symbols ($\times$, right $y$ axis) are fluxes that passed quality control.

is likely advection of polluted air offshore, and the $SO_2$ levels measured during onshore flow may be elevated compared to marine air from the open ocean. The average air–sea temperature differential during the study was $0.56 \pm 1.55\,°C$ with a range from $-3.5$ to $2.7\,°C$, with positive values indicating a warmer ocean than atmosphere. Occasionally air–sea temperature differentials exhibited diurnal variability which reflected the changes in air temperatures. Starting on DOY 114, seawater temperatures warmed and were significantly warmer than air temperatures for the remaining 3 days of the study.

## 4.2 Air–sea differences and fluxes

All the observed $SO_2$ fluxes were from the atmosphere to the ocean surface (negative by convention) and ranged from 0 to $-65\,pmol\,m^{-2}\,s^{-1}$, with the largest fluxes observed at the beginning and end of the deployment associated with high $SO_2$ levels and high wind speeds, respectively (Fig. 1). All observed water vapor and sensible heat fluxes passing quality control were upward, which was consistent with the positive (from the ocean to the atmosphere) thermodynamic gradient for the duration of the study. The warm seawater temperatures combined with the high winds and cold temperatures on the last 2 days of the study resulted in large $H_2O$ and heat fluxes.

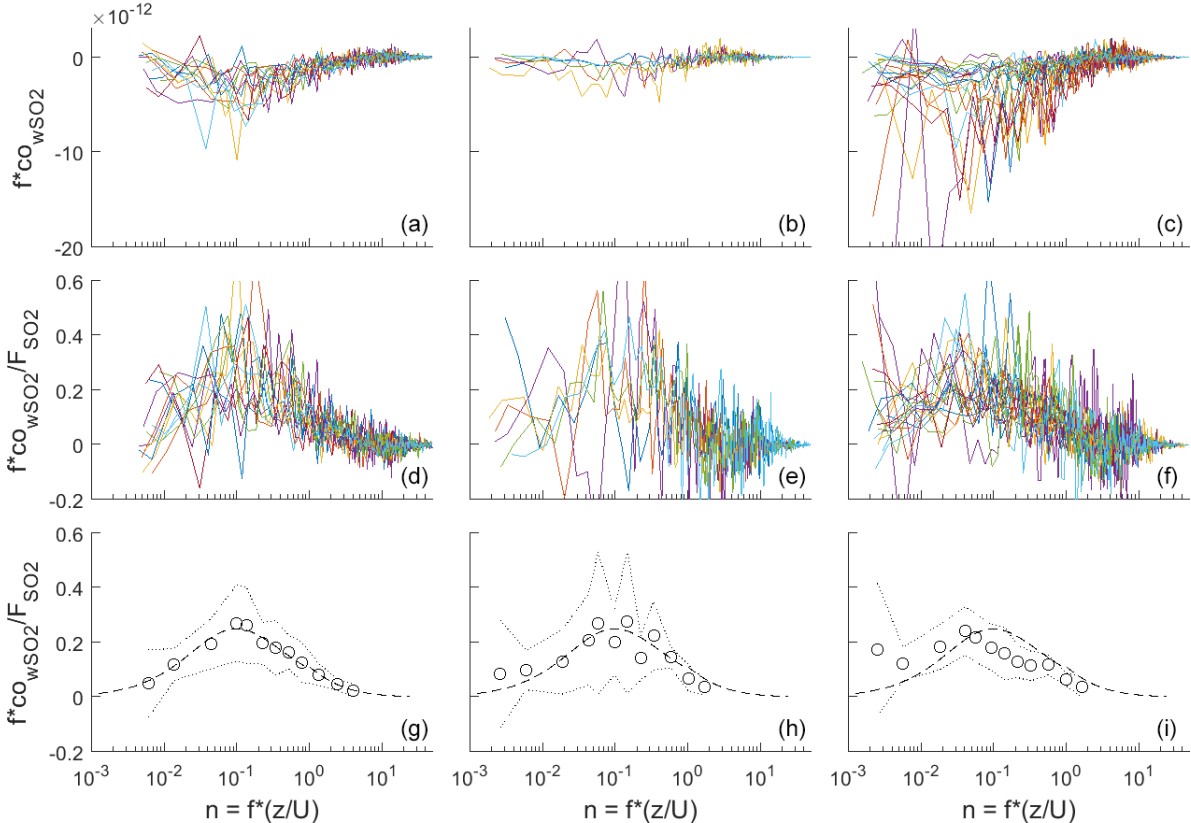

**Figure 2.** Frequency-weighted co-spectra of vertical wind and $SO_2$ concentration for flux intervals collected at Scripps Pier during three time periods. **(a, d, g)** DOY 96–102; **(b, e, h)** DOY 104–109; **(c, f, i)** DOY 114–117. **(a–c)** Individual co-spectra for 13 min flux intervals; **(d–f)** same as top except co-spectra have been normalized to the average flux during the interval. **(g–i)** Bin-averages of the flux-normalized co-spectra (circles), $\pm$ 1 standard deviation (dotted line), and idealized co-spectral shape from Kaimal et al. (1972) (dashed line).

Frequency-weighted co-spectra of vertical wind and $SO_2$ are shown in Fig. 2. Fluxes measured during DOY 114–117 were significantly larger than those measured during the rest of the campaign because of the strong winds and large air–sea temperature differences observed during that period (Fig. 1). The co-spectra measured at Scripps Pier for all parameters were similar in shape to the characteristic boundary layer co-spectral shapes defined by Kaimal et al. (1972).

### 4.3 Transfer velocities

The wind speed dependence of $k_{mom}$ observed in this study was significantly greater than predicted using the open ocean parameterization from the NOAA COARE (Fairall et al., 2000) (Fig. 3). The relationship between wind speed and surface roughness can vary significantly between the open ocean and coastal environments because of bottom-generated turbulence, as well as other influences related to fetch, tidal currents, surfactants, and wave properties (Smith, 1988; Brown et al., 2013; Geernaert et al., 1986). Thus, the turbulent properties of the atmospheric surface layer in coastal environments are not well described by wind speed alone. To account

for such effects, we examined the relationship between transfer velocities and both wind speed and friction velocity ($u_*$) (Fig. 4).

The transfer velocities measured for water vapor, sensible heat, and $SO_2$ ($k_{H_2O}$, $k_{SH}$, $k_{SO_2}$) were all positively correlated with friction velocity (Fig. 4, Table 1). $k_{mom}$ was significantly higher than the scalar parameters and $k_{SO_2}$ was lower than $k_{H_2O}$ and $k_{SH}$. The regressions against friction velocity utilize slightly different data sets in each case because these regressions utilize flux measurement intervals that passed quality control for both the scalar parameter (water vapor, sensible heat, $SO_2$) and for momentum flux. This means that the data sets used for the various parameters were not identical either in terms of the number of flux intervals or the physical conditions under which they were collected, i.e., temperature, wind speed, atmospheric stability, sea state, etc. Ideally, the comparison of transfer velocities would be carried out using intervals for which all four of the parameters passed quality control. However, given the limited data set, this constraint reduced the available data to an unacceptable degree. As an alternative, we also compared the gas transfer velocities to each other by computing two-way linear re-

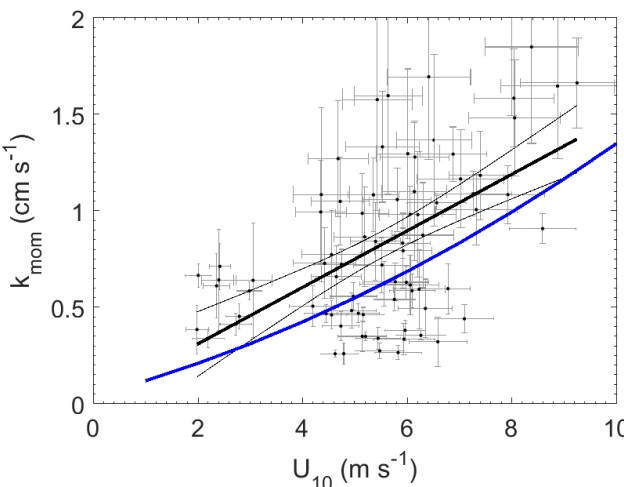

**Figure 3.** Momentum transfer velocities measured at Scripps Pier as a function of wind speed with linear least squares regression and 95 % confidence intervals (black). Blue line – COAREG parameterization of Fairall et al. (2000).

gressions between pairs of simultaneously measured transfer velocities (Fig. 5, Table 2). This analysis was in general agreement with the $k$ vs. $u_*$ analysis described earlier and showed $k_{SO_2} < k_{H_2O}$, $k_{SO_2} < k_{SH}$ and no significant difference between $k_{SH}$ and $k_{H_2O}$. Momentum transfer velocities were significantly larger than all the scalar transfer velocities. The comparison of transfer velocities from simultaneous intervals is a more robust approach to observing differences in transfer velocities.

## 5   Discussion

This study demonstrates the successful measurement of $SO_2$ deposition to the sea surface using eddy covariance, with (1) co-spectra exhibiting a similar shape to water vapor and sensible heat and (2) a linear relationship between transfer velocities and wind speed or friction velocity. Virtually all of the $SO_2$ co-spectra indicated that the direction of flux was from air to sea, even during periods of very low atmospheric $SO_2$. This confirms the assumption that seawater $SO_2$ concentrations are highly undersaturated with respect to atmospheric $SO_2$. In general, we expect measurements of $k_{SO_2}$ to be of higher precision than those of water vapor and sensible heat because (1) the $SO_2$ in seawater is negligible, so the air–sea concentration gradient is equal to the bulk atmospheric concentration, eliminating the need for a water-side measurement; and (2) the $SO_2$ flux and atmospheric concentration are determined simultaneously using a single sensor with a linear response, so the absolute calibration of the sensor does not influence the measured gas transfer velocity. These are advantages compared to the measurement of transfer velocities for water vapor or sensible heat, which require both air-side

and water-side measurements in order to quantify the air–sea concentration or temperature difference. The transfer velocities for $SO_2$ had significantly less scatter compared to the water vapor and sensible heat transfer velocities at high wind speeds (Fig. 4).

Faloona et al. (2009) reported airborne eddy covariance measurements of $SO_2$ deposition over the equatorial Pacific. The data from their lowest flight altitude of 30 m should be comparable to the data from this study. We made this comparison as a function of $u_*$ rather than wind speed to account for the differences in sea surface roughness between the coastal and open ocean environments. The $SO_2$ transfer velocities reported by Faloona et al. (2009) were roughly half those observed at Scripps over a similar range of wind stress (Fig. 6, Table 3). This difference is considerably larger than expected from the scatter in the data or estimated uncertainties in the flux measurements. Further investigation is needed in order to determine whether a systematic difference exists in $SO_2$ deposition to coastal vs. open ocean waters and, if so, what the cause might be.

A few studies of direct air–sea exchange of highly soluble organic compounds have also been carried out. Fluxes of acetone to the Pacific Ocean were reported by Marandino et al. (2005) and methanol fluxes to the Atlantic Ocean were reported by Yang et al. (2013). Surprisingly, the direction and/or magnitude of air–sea fluxes observed in those studies were not consistent with observed air–sea concentration differences based on bulk air and seawater measurements. Both studies speculated that this was due to near-surface water-side gradients, because assuming a zero sea surface concentration gave reasonable gas transfer velocities with linear wind speed dependence. For acetone, the resulting gas transfer velocities were considerably lower than those observed in this study (Fig. 6, Table 3). For methanol, the gas transfer velocities were similar to this study, but with a slightly stronger dependence on wind stress. The anomalous behavior of acetone and methanol is generally thought to be related to near-surface biological or photochemical processes. The presumed near-surface gradients are problematic in that they require strong localized production and loss processes and have not yet been observed in the field. Given the uncertainty introduced by these inferred gradients, more detailed analysis of the similarities and differences in the data seem unwarranted.

One of the goals of this study was to compare observations of air-side-controlled gas transfer velocities to model parameterizations. The COAREG air–sea gas transfer model (Fairall et al., 2000, 2011) utilizes the open ocean COARE parameterization of friction velocity, based on wind speed and stability (Fairall et al., 1996). As a result, COAREG substantially underestimates the observed transfer velocities for this nearshore coastal site. As noted earlier, momentum transfer coefficients at Scripps Pier were elevated compared to those typically encountered under open ocean conditions. COAREG yields much better agreement with the field data

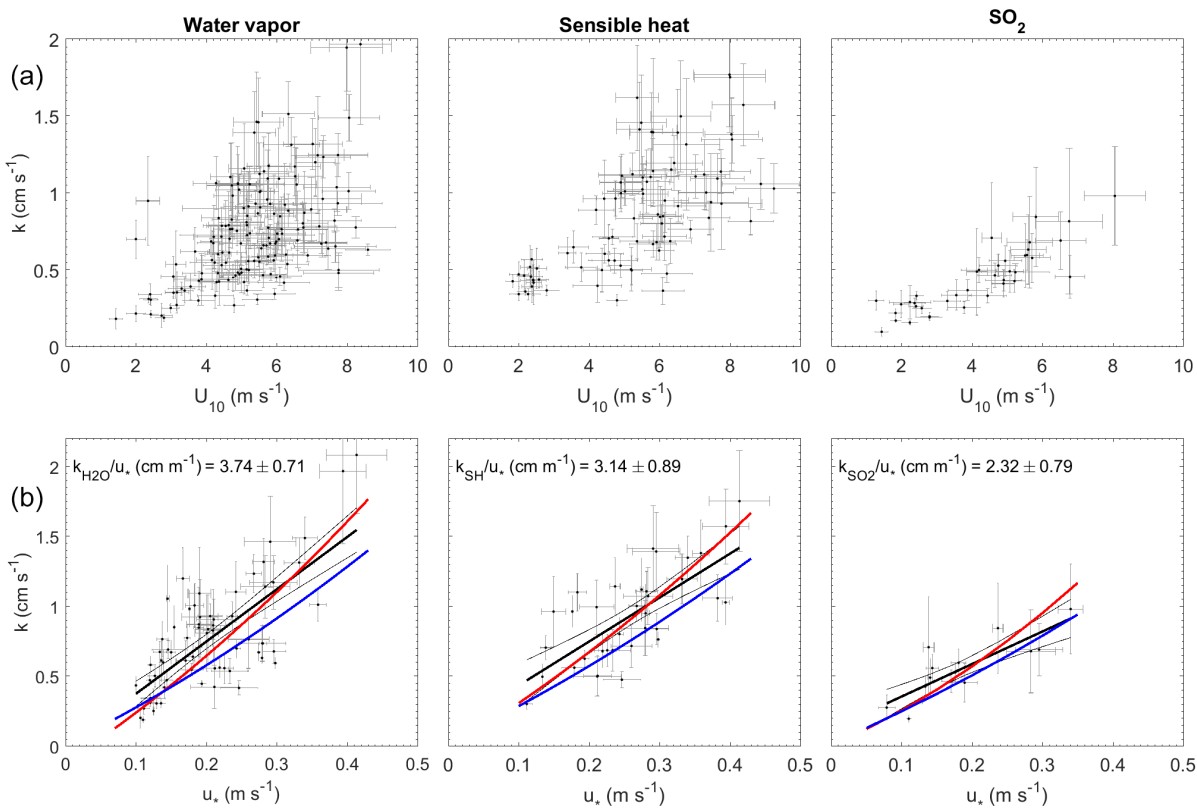

**Figure 4.** Transfer velocities measured at Scripps Pier as a function of wind and friction velocity. **(a)** Water vapor, sensible heat, and $SO_2$ as a function of $U_{10}$ (black dots). **(b)** Water vapor, sensible heat, and $SO_2$ as a function of $u_*$ with linear least squares regressions and 95 % confidence intervals (black dots and black line). Red lines are a second-order least squares regression of transfer velocities computed with the COAREG parameterization using measured drag coefficients (Fairall et al., 2000, 2011). Blue lines are transfer velocities computed with the COAREG parameterization, allowing the model to calculate friction velocities and drag coefficients.

**Table 1.** Two-way regression of transfer velocities against friction velocity ($k/u_*$).

| Parameter | Regression slope $\pm$ CI ($\alpha = .05$) (cm m$^{-1}$) | Number of observations |
|---|---|---|
| Water vapor ($k_{H_2O}/u_*$) | $3.74 \pm 0.71$ | 69 |
| Sensible heat ($k_{SH}/u_*$) | $3.14 \pm 0.89$ | 36 |
| Sulfur dioxide ($k_{SO_2}/u_*$) | $2.32 \pm 0.79$ | 15 |
| Momentum ($k_{mom}/u_*$) | $5.06 \pm 0.40$ | 80 |

when drag coefficients based on the measured momentum fluxes were used (Figs. 4, 6). In this study, the momentum transfer velocity was significantly (roughly 50 %) larger than the transfer velocities of $SO_2$, $H_2O$, and sensible heat observed under simultaneous or similar conditions. This is reasonable, given that momentum can be transferred across the air–sea interface via both viscous stress (analogous to diffusion of mass or heat) and by pressure forces for which there is no analog in mass transfer.

Differences between the gas transfer velocities of $SO_2$, $H_2O$, and sensible heat should reflect the role of molecular diffusivity in the viscous layer adjacent to the sea surface.

The diffusivity of $SO_2$ in air is roughly half that of $H_2O$ or sensible heat (Table 4). Comparing the relative magnitudes of $k_{H_2O}$, $k_{SH}$, and $k_{SO_2}$ is therefore a good test for the ability of gas transfer models to partition resistance between turbulence and diffusion. Using the drag coefficients based on the field data, COAREG gives $k_{SO_2}/k_{H_2O} = 0.82$. Using the average $k/u_*$ of the field observations (Fig. 4) gives

$$\frac{k_{SO_2}/u_*}{k_{H_2O}/u_*} = \frac{2.32 \pm 0.79}{3.74 \pm 0.71} = 0.62 \pm 0.24. \qquad (25)$$

The pairwise analysis of simultaneous measurements gives a ratio of $k_{SO_2}/k_{H_2O}$ of $0.52 \pm 0.14$. Thus, the field obser-

**Table 2.** Pairwise regression of transfer velocities using simultaneously measured data from Figs. 3 and 4.

| Parameter | Regression slope $\pm$ CI ($\alpha = .05$) | Number of data points |
|---|---|---|
| Sulfur dioxide vs. water vapor ($k_{SO_2}$ vs. $k_{H_2O}$) | $0.52 \pm 0.14$ | 26 |
| Sulfur dioxide vs. sensible heat ($k_{SO_2}$ vs. $k_{SH}$) | $0.64 \pm 0.15$ | 20 |
| Water vapor vs. sensible heat ($k_{H_2O}$ vs. $k_{SH}$) | $1.17 \pm 0.15$ | 64 |
| Sulfur dioxide vs. momentum ($k_{SO_2}$ vs. $k_{mom}$) | $0.40 \pm 0.27$ | 15 |
| Water vapor vs. momentum ($k_{H_2O}$ vs. $k_{mom}$) | $0.82 \pm 0.15$ | 69 |
| Sensible heat vs. momentum ($k_{SH}$ vs. $k_{mom}$) | $0.72 \pm 0.13$ | 36 |

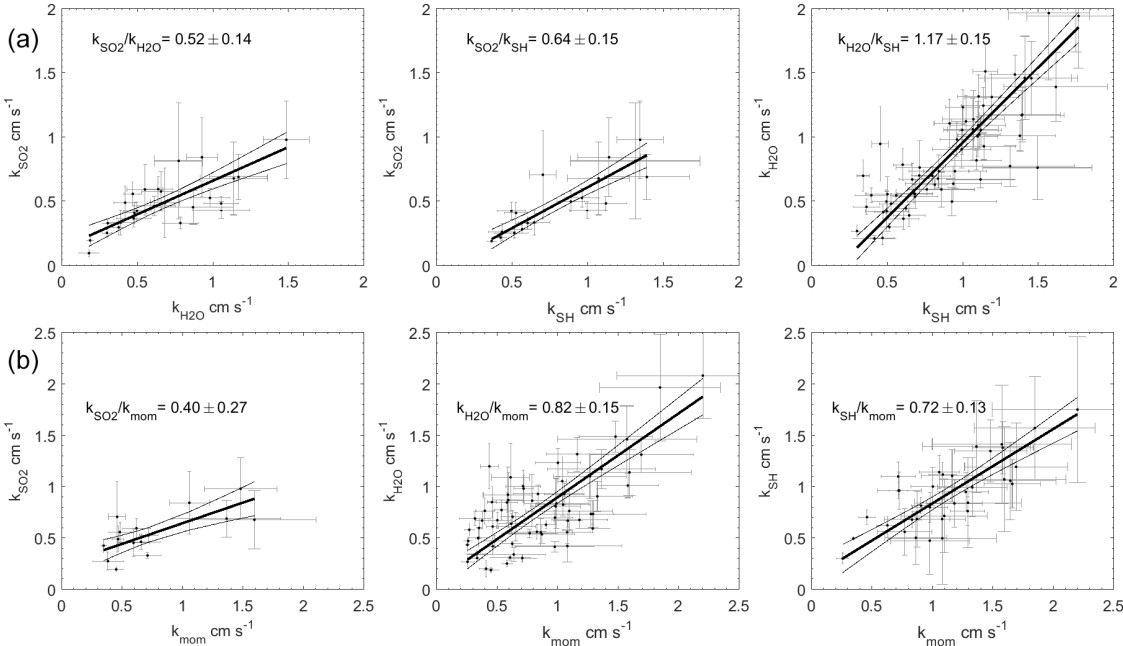

**Figure 5.** Two-way regressions of transfer velocities measured at Scripps Pier. **(a)** Water vapor, sensible heat, and SO$_2$ against each other. **(b)** SO$_2$, water vapor, and sensible heat regressed against momentum. The 95 % confidence intervals are shown.

vations and model qualitatively agree that the resistance to SO$_2$ transfer is greater than that of H$_2$O. Quantitatively, the COAREG result is just within the 95 % confidence interval of the $k/u_*$ result, but outside the uncertainty range of the pairwise comparison. For $k_{SO_2}/k_{SH}$ the result is similar, with better agreement between observations and model. COAREG predicts a ratio of 0.85 while the field data yield $0.74 \pm 0.33$ from the ratio of average $k/u_*$ and $0.64 \pm 0.15$ from the pairwise analysis. Finally, for $k_{H_2O}/k_{SH}$ COAREG predicts a ratio of 1.03. This agrees very well with the field observations, which give ratios of $1.19 \pm 0.41$ from the average $k/u_*$ and $1.17 \pm 0.15$ from the pairwise analysis. The model–data agreement for $k_{H_2O}/k_{SH}$ is not surprising because their $Sc$ numbers are almost identical. Consequently, the ratio calculated by COAREG should not be sensitive to either the partitioning between turbulent and diffusive resistance or to the parameterization of diffusive resistance.

The field data suggest that the resistance to gas transfer of SO$_2$ is larger than expected from COAREG. This could indicate that COAREG underestimates diffusive resistance or it could indicate some additional unknown source of resistance, such as a surface resistance. It seems unlikely, though not impossible, that surface resistance associated with the sea surface microlayer would influence only SO$_2$ and not H$_2$O, but as noted earlier, the properties of the sea surface microlayer are not well known. We can estimate the magnitude of this anomalous resistance using the field data and COAREG as follows:

$$r_{\text{total\_H}_2\text{O}} = r_{\text{turb}} + r_{\text{diff\_H}_2\text{O}} = r_{\text{H}_2\text{O\_COAREG}}, \tag{26}$$

$$r_{\text{total\_SO}_2} = r_{\text{turb}} + r_{\text{diff\_SO}_2} + r_{\text{anom\_SO}_2}$$

$$= r_{\text{SO}_2\text{\_COAREG}} + r_{\text{anom\_SO}_2}, \tag{27}$$

$$\frac{r_{\text{SO}_2\text{\_COAREG}}}{r_{\text{H}_2\text{O\_COAREG}}} = 1.18. \tag{28}$$

**Table 3.** Slopes and intercepts of regressions to $k$ vs. $u_*$ shown in Fig. 6. 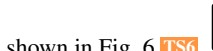

| References | Gas | Slope $\pm$ 95 %CI | Intercept $\pm$ 95 %CI |
|---|---|---|---|
| This study | SO$_2$ | $2.74 \pm 0.62$ | $0.07 \pm 0.11$ |
| Faloona et al. (2009) | SO$_2$ | $1.20 \pm 0.50$ | $0.10 \pm 0.12$ |
| Yang et al. (2013) | methanol | $3.82 \pm 0.29$ | $-0.22 \pm 0.08$ |
| Marandino et al. (2005) | acetone | $1.28 \pm 0.34$ | $0.05 \pm 0.07$ |

**Table 4.** Diffusion coefficients and Schmidt numbers for H$_2$O, sensible heat, and SO$_2$ in air, as calculated according to Fuller et al. (1966) and Hilsenrath (1960).

| Parameters | H$_2$O | Sensible heat | SO$_2$ |
|---|---|---|---|
| Diffusion coefficient in air (298 K; cm$^2$ s$^{-1}$) | 0.25 | 0.22 | 0.13 |
| $Sc$ number ($Sc$; 298 K) | 0.61 | 0.69 | 1.19 |

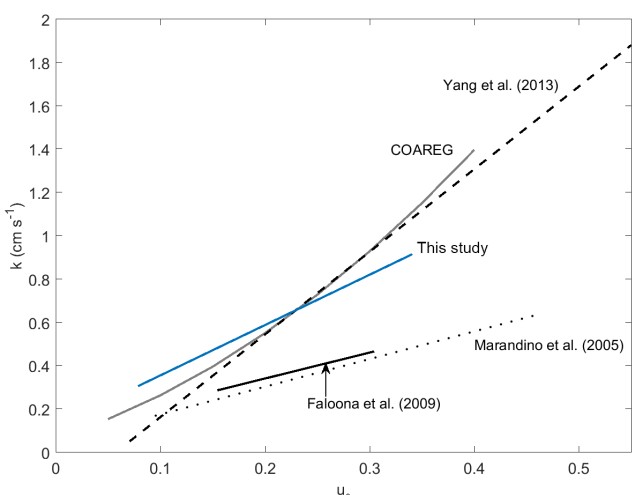

**Figure 6.** Gas transfer velocities as a function of friction velocity for this study and prior measurements of air–sea exchange of highly soluble, air-side-controlled gases from Yang et al. (2013), Faloona et al. (2009), Marandino et al. (2005), and this study. The grey line is the COAREG model calculated with the drag coefficients measured during this study, using the $Sc$ number of SO$_2$.

The $k/u_*$ slopes of the field data give

$$\frac{r_{\text{total\_SO}_2}}{r_{\text{total\_H}_2\text{O}}} = \frac{k_{\text{H}_2\text{O}}/u_*}{k_{\text{SO}_2}/u_*} = 1.61 \pm 0.63. \qquad (29)$$

Solving these equations simultaneously yields

$$r_{\text{anom}}/r_{\text{total}_{\text{SO}_2}} = 0.26 \pm 0.29. \qquad (30)$$

The analysis using the pairwise data gives

$$r_{\text{anom}}/r_{\text{total}_{\text{SO}_2}} = 0.38 \pm 0.17. \qquad (31)$$

In other words, the field data allow for additional resistance for SO$_2$ comprising 25 %–38 % of the total air-side SO$_2$ re-

sistance. However, given the limited data set and the uncertainties associated with the regressions, it seems premature to conclude that such anomalous resistance exists or to speculate on its origin. It does seem likely that, with further work, measurements such as these can provide useful constraints on air–sea gas transfer models.

## 6 Conclusions

This study demonstrated successful measurement of atmospheric deposition of sulfur dioxide to the sea surface by eddy covariance. The high effective solubility and negligible seawater concentrations make SO$_2$ a useful tracer for studying the processes controlling air-side resistance to air–sea gas transfer. The deposition velocities found in this study are in reasonable agreement with bulk parameterizations in current use. The data from this study show that SO$_2$ transfer velocities are lower than those of momentum and water vapor, in qualitative agreement with gas transfer theory. The measurement of air–sea SO$_2$ fluxes provides the opportunity to compare the transfer rates of air-side-controlled substances with different molecular diffusivities. This study was limited in terms of both the amount of data collected and the range of environmental conditions sampled. Further studies conducted on the open ocean, covering a wider range of wind speeds, sea state, and air–water temperature differences, could make a significant contribution to our understanding of the deposition of highly soluble gases to the oceans.

*Data availability.* Data used in this study are available at https://doi.org/10.7280/D16M24. This is a data repository maintained by the University of California.

**The Supplement related to this article is available online at: https://doi.org/10.5194/acp-18-1-2018-supplement**

*Author contributions.* JGP carried out the field measurements, processed the data, and contributed to the writing of the manuscript. WDB contributed to the field setup, data interpretation, and writing and editing of the manuscript. ESS contributed to the instrumentation development, experimental design, data interpretation, and writing and editing of the manuscript.

*Competing interests.* The authors declare that they have no conflict of interest.

*Acknowledgements.* We wish to thank Christian McDonald and the Scripps Institute of Oceanography for use of the Scripps Pier and to Eric Terrill of the Scripps Coastal Observing Research and Development Center for sea surface temperature data. We especially wish to thank Keqi Tang at Pacific Northwest National Laboratory for assistance in the design and construction of the ion funnel used in this study, as well as Scott Miller of SUNY Albany for scientific discussions. Cyril McCormick of the UCI Instrumentation Development Facility provided support in the field and laboratory. Support for this research was provided by NASA (grant NNX15AF31G) and the NSF IR/D program.

Edited by: Timothy Bertram
Reviewed by: Mingxi Yang and Byron Blomquist

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

**Remarks from the language copy-editor**

CE1    Please be more specific as to what needs to be changed here.

**Remarks from the typesetter**

TS1    Please confirm the order and numbering of the reactions and equations.
TS2    Please note that all changes to numbers and units (and therefore the content of the paper) would have to be accepted by the editor first. If you insist on inserting these changes, please send a statement explaining why they are necessary. Thank you.
TS3    Please check all in text references to equations and reactions to check if they are still correct.
TS4    Please confirm this is the sign you asked for.
TS5    Please confirm the change to Eq. 20.
TS6    Please check all references to Tables and confirm them.
TS7    Please provide a reference including creators, title, and date of last access.
TS8    Please provide volume.