# Peer review of "Eddy flux measurements of sulfur dioxide deposition to the sea surface"

_Atmospheric Chemistry and Physics, 2018_

## Referee Comment (RC1) · M. Yang (Referee) · 3 Jul 2018

Review for "Eddy flux measurements of sulfur dioxide deposition to the sea surface"

This is a nice paper describing direct measurements of air-sea SO2 transfer, which were made at a coastal location. This is a highly challenging measurement, as evidenced by the very few previous data for comparison. The measurements reported here appear to be sound and the data processing/filtering procedure reasonable (though slightly incomplete, see detailed comments). The fact that the results are largely consistent with existent theory doesn't reduce the value of this work. As the authors mentioned, these measurements were made over a very limited range of environmental conditions. Further observations over a much wider range of conditions

have the potential to not only improve our understanding of air-sea SO2 transfer, but also the transfer of many other highly soluble gases as well as heat. In my opinion the paper can be published after some technical corrections.

Specific comments: P1, line 18. Please give references and indicate ranges. And note that anthropogenic SO2 emissions are changing globally, with for example decreasing trends in Western Europe and North America.

P2. Line 1-3. Please give references. Yang et al (PNAS 2013, ACP 2014, GTWS 2016) measured the air-sea transfer of methanol and compared its rate to those of momentum and sensible heat. A very similar method of analysis is used here.

P. 2 Line 6-7. The previous sentence just said that the Faloona et al's measurements were in the MBL. Also, it should be 2010, not 2009.

P. 5. Line 12. What's the tidal range at this site? If significant, it'd alter the measurement height above the mean sea level and so the extent of the flux footprint.

P. 7 line 12. How is the SO2 blank measurement (what I assume the authors meant by 'system blanks') made? Since Vd = -flux/[C], any error in mean [C] due to uncertain blank correction will propagate to Vd. To test, the authors can plot Vd vs. u* and color-code it by [C]. Is there any pattern?

P. 7 line 20. What's the typical tilt angle, i.e. the angle between the horizontal and the streamline?

P. 7, line 30. The authors show later that the mean Cd measured at this location is significantly greater than what the COARE model predicts. If so, U10 computed from the COARE model (assuming open ocean Cd) will be in error. In theory, to get a more accurate U10 it's probably better to use an iterative approach to estimate U10 from the measured Cd. Though the difference might not be very big in this case since the measurement height is already at ∼10 m above sea level.

p. 7 line 32. Last sentence: 'sensible heat' is left out

p. 8. Equation 11. The Licor7500 measures mass concentrations rather than the mixing ratios. Thus a 'Webb' correction for air density fluctuation is required for water vapor flux (probably not a big correction). Has this been applied?

p. 8 Equation 13. Is T here the sonic temperature or the air temperature? The sonic temperature, approximately equal to virtual temperature, is affected by humidity. Thus one needs to apply a latent heat correction to the sonic heat flux to get sensible heat flux. This probably isn't a big correction, and can be achieved by:

a) apply a high frequency (e.g. >=5 Hz, if available) humidity correction to the raw sonic temperature data, or b) use the actual latent heat flux (or bulk latent heat flux) to correct the sonic heat flux

P. 8, Equation 17. Typically a lapse rate correction is applied to the measured air temperature in the calculation of deltaT. What's the height of the mean air temperature sensor?

More generally, airside transfer is dependent on atmospheric stability. It is typical to convert the airside transfer velocities to neutral transfer velocities. This doesn't affect the authors pair-wise comparisons (e.g. kSO2 vs kSH), but does affect the kSO2 vs u* relationship, for example.

P. 9, line 1. How did the authors arrive at a cut off frequency of 1.5 Hz? What is the instrument response time?

p. 9, line 16. Have the authors estimated the flux detection limit for their SO2 system? SO2 over the open ocean (especially Southern Hemisphere) is only typically a few tens of ppt.

p. 9, line 18. Reasoning and reference for excluding data with Z/L > 0.07?

P. 9. Line 19. What were the thresholds used for excluding SO2 ship spikes?

p. 10, line 25. There is very large variability in the kmom vs. U10 relationship here.

Typically over the open ocean, the relative standard deviation of u* decreases with increasing wind speed. Could some of the variability here be due to tidal height changes or the wind direction being on the edge of the acceptable sea-sector? Also, see my comment about U10 earlier.

P. 11 line 13. At the beginning of this section, I think the authors should first compare their kSO2 vs. u* relationship to a) the only previous measurements of kSO2 (Faloona et al. J. Atmos Chem 2010), and b) kmethanol from Yang et al. GTWS 2016.

p. 11, line 18. The authors' assertion that kSO2 is more precise than kSH and kH2O appears to be backed up by their Fig. 4. With increasing wind speed, the scatter in kSO2 only increases marginally (the relative standard deviation probably decreases). In contrast, the scatter in kSH and kH2O increase substantially with wind speed.

p. 11, line 25. Technically the name COAREG started with Fairall et al. 2011 JGR (the gas transfer version), not Fairall et al. 2000.

p. 11, line 28-30. It looks like the author substituted computed Cd with the mean Cd v. U10 relationship from the measurements? Would the authors be able to explain more of the variability in the other k data if they prescribe the model with the measured Cd on a point-by-point basis?

p. 12, line 16. The authors should specify that these are airside Schmidt numbers, which are largely temperature independent. Also, how does the Sc_SO2 here compare with more contemporary predictions (e.g. from Johnson 2000 Ocean Science)?

P. 12 line 19-20. Quoted uncertainties here for the Schmidt number exponents are very large. Are they derived from a regression of rdiffH2O vs. rdiffSO2, etc? There seems to be a lot of variability in the kmom data, which isn't as apparent in the kSO2 data. Is subtracting such a noisy rturb (from kmom) the cause for the poor regression results?

p. 12 line 22. More appropriate references than Jaehne et al. 1987 are specific studies of airside transfer, including Hicks et al. 1986 as well as the earlier Liss et al. papers.

[Figure]

p. 23, Fig 4. Show units for slope. Also, typo in caption: 'computed' instead of 'computer'

Finally, I personally find 'higher/lower' to be more suitable adjectives for transfer velocity than 'bigger/smaller'.

The authors can contact me directly for further questions if they wish.
* * *

---

## Short Comment (SC1) · 17 Jul 2018

This is a nice paper on a topic of broad relevance to air-sea gas transfer. The paper is generally well written. The introduction and method sections are complete and clear. I will make some general comments on the analysis and conclusions, followed by a list of minor edits and suggestions.

This paper deals with the deposition of a reactive gas, sulfur dioxide, to the ocean surface. Ionic equilibria and oxidation reactions in the aqueous phase (discussed in Sec 2.2) are sufficiently rapid that the equilibrium water-side concentration of $SO_2$ is quite small, and thus we expect physical processes on the air-side will determine the deposition rate constant. But, as mentioned at the end of 2.2.1, we expect surface effects

(mass accommodation) may also be important, and this additional resistance to mass transfer is distinct from either turbulent diffusion in the bulk air or molecular diffusion in the surface microlayer. The physical diffusive mechanisms are fairly well understood. The surface effects are not. The magnitude of this barrier probably depends on surface microlayer characteristics, and for the ocean it could be significantly different from the resistance at the surface of small droplets implied by the Worsnop et al 1989 laboratory data.

Given the excellent introduction I was a bit disappointed that the discussion and conclusions did not try to address the surface resistance, but rather seem to lump it into a diffusion factor which doesn't lead to significant insights.

For gases like SO2, I think the airside resistances are better represented by

r_total = r_turbulence + r_diffusion + r_surface

It seems to me that you should be able to estimate r_surface for SO2 as the difference between r_total for SO2 and r_total for water vapor, since there is no surface resistance effect for water. You can also compare r_total for SO2 with r_a from COAREG, which is a stability-corrected estimate of the turbulent and molecular diffusion resistances but does not consider a surface resistance.

This sort of analysis could lead to improvements in the COAREG parameterization through the introduction of an additional species-dependent constant in the formulation of r_a. Even if the data available from this study are insufficient for a statistically significant determination of the surface resistance term, the exercise provides useful guidance for future investigations.

Other more minor suggestions and comments:

P2 Eq. 1: You could point out that the delta C term defined this way means a downward flux is positive, which is typical in literature dealing with deposition to the surface, but opposite the general convention for gas transfer where the upward flux is positive.

P2 Line 20: define solubility as Cw_o/Ca_o to emphasize this is the equilibrium concentration ratio? Personally, I would avoid the term 'Ostwald coefficient' since this is subject to several slightly different and potentially confusing definitions (see R. Battino, Fluid Phase Equilibria, 15, 231-240, 1984) and just call it the dimensionless (liq/gas) solubility.

P4 line 11: 'This is sufficiently small. . .' ?

Sec 3.2: Were blank measurements conducted and if so, how? In previous work with this method we have used a coil of HCl-washed copper tube to remove SO2 from the sample stream and determine the background signals at m/z 112 and m/z 114. Admittedly, this is not a perfect blank, because removing one reactant from the air sample perturbs the ion-molecule equilibria in the source, such that the background signal you measure in the absence of SO2 may not be exactly the same as the background when SO2 is present. If the concentration of SO2 (ambient + internal standard) is small compared to the CO2 and ozone concentrations, however, this consideration should be minor.

P7, eq 9: F is being used for both flux and flow which is confusing. Choose another variable in this equation to represent gas flow. . .

P7, line 30: The correct references for the COARE model are Fairall et al. 1996, Fairall et al. 2003 and Edson et al. 2013. Fairall et al. 2000 deals specifically with gas transfer.

P8 line eq 12: You might mention that, Fmom is more commonly called the Reynolds stress (tau).

Sec 5: See comments above. Also, you could mention that a linear wind speed dependence is expected for very soluble gases and has been demonstrated in other studies (i.e. little or no bubble enhancement to k from breaking waves). In comparisons with the physical model I would just use k_a from COAREG and ignore k_b which should

not be important and in any case is the more uncertain parameter.

P11 line 25: The other reference for updates to COAREG is Fairall et al., 2011.

---

## Author Comment (AC1) · 13 Sep 2018

Author comments in response to reviewer comments.

The authors wish to thank both reviewers for their insightful and detailed reviews. The manuscript has been revised in accordance with their comments. Our responses to specific comments are given below.

Author responses to comments by Reviewer 1:

RC1: P1, line 18. Please give references and indicate ranges. And note that anthropogenic SO2 emissions are changing globally, with for example decreasing trends in Western Europe and North America.

[Figure]

Author response: References have been added and ranges are indicated.

RC1: P2. Line 1-3. Please give references. Yang et al (PNAS 2013, ACP 2014, GTWS 2016) measured the air-sea transfer of methanol and compared its rate to those of momentum and sensible heat. A very similar method of analysis is used here.

Author response: References added.

RC1: P. 2 Line 6-7. The previous sentence just said that the Faloona et al's measurements were in the MBL. Also, it should be 2010, not 2009.

Author response: This text has been revised as follows. "Faloona et al. (2009) reported air/sea eddy covariance surface fluxes for $SO_2$ using a fast-response chemical ionization mass spectrometric technique developed by Bandy et al. (2002). To our knowledge these are the only previous eddy covariance measurement of $SO_2$ surface fluxes over the ocean." There appears to be some confusion about the date of this journal article. The doi links to a pdf that indicates the paper was accepted for publication and copyrighted in 2010, but the article appears in the 2009 issue of the Journal of Atmospheric Chemistry. We are inquiring with the publisher to resolve this issue.

RC1: P. 5. Line 12. What's the tidal range at this site? If significant, it'd alter the measurement height above the mean sea level and so the extent of the flux footprint.

Author response: The tidal range over the course of the experiment was 1.69m. The paper was revised as follows: "The sensing regions of the eddy covariance flux package and the air intake for $SO_2$ detection were located approximately 10 m above the sea surface. The sensor height was corrected for changes in tidal range during the experiment."

RC1: P. 7 line 12. How is the SO2 blank measurement (what I assume the authors meant by 'system blanks') made? Since Vd = -flux/[C], any error in mean [C] due to uncertain blank correction will propagate to Vd. To test, the authors can plot Vd vs. u* and color-code it by [C]. Is there any pattern?

Author response: The text was revised to clarify the procedure for blank measurements, as follows: "...where $S_{112}$ and $S_{114}$ are blank corrected mass spectrometer signals, $f_{std}$ and $f_{total}$ are the gas flow rates of the isotopic standard and inlet and Xtank is the molar mixing ratio of $^{34}SO_2$ in the compressed cylinder. Because the air stream was dried in the inlet tube prior to analysis, $X_{SO2}$ represents the mixing ratio of $SO_2$ in dry air. Blanks were obtained by sampling air through a carbonate-impregnated filter to quantitatively remove ambient $SO_2$. Whatman 41 filters for this purpose were soaked in 1% sodium carbonate solution and dried prior to use."

We made the suggested plot of $V_d$ vs. $u_*$, color-coded by [C]. No relationship was observed between [C] and the $SO_2$ transfer velocities.

RC1: P. 7 line 20. What's the typical tilt angle, i.e. the angle between the horizontal and the streamline?

Author response: The average tilt angle was 1.3 degrees. The manuscript was changed to include this information as follows: "...rotating the 3-D winds for each flux interval into the frame of reference of the mean winds and to account for tilt in the sonic anemometer ($1.3°$)..."

RC1: P. 7, line 30. The authors show later that the mean Cd measured at this location is significantly greater than what the COARE model predicts. If so, U10 computed from the COARE model (assuming open ocean Cd) will be in error. In theory, to get a more accurate U10 it's probably better to use an iterative approach to estimate U10 from the measured Cd. Though the difference might not be very big in this case since the measurement height is already at 10 m above sea level.

Author response: We agree. We made the suggested calculations for $U_{10}$. As noted, the difference is quite small, with an average of only about 0.1% difference from the uncorrected wind speeds. Figures 1, 3, and 4 and the text in section 3.3 were updated to reflect the new wind speed calculations.

RC1: p. 7 line 32. Last sentence: 'sensible heat' is left out

Author response: Corrected

RC1: p. 8. Equation 11. The Licor7500 measures mass concentrations rather than the mixing ratios. Thus a 'Webb' correction for air density fluctuation is required for water vapor flux (probably not a big correction). Has this been applied?

Author response: This Webb correction was applied. The text was revised to explicitly indicate that the correction was done as follows: "Water vapor concentrations measured by the LICOR were corrected to account for air density fluctuations and converted to concentration (mol m$^{-3}$)."

RC1: p. 8 Equation 13. Is T here the sonic temperature or the air temperature? The sonic temperature, approximately equal to virtual temperature, is affected by humidity. Thus one needs to apply a latent heat correction to the sonic heat flux to get sensible heat flux. This probably isn't a big correction, and can be achieved by: a) apply a high frequency (e.g. >=5 Hz, if available) humidity correction to the raw sonic temperature data, or b) use the actual latent heat flux (or bulk latent heat flux) to correct the sonic heat flux

Author response: The T in equation 13 is air temperature after applying a high frequency humidity correction to the raw sonic temperature data. The manuscript was revised as follows to indicate this: "...T is the humidity-corrected air temperature ..."

RC1: P. 8, Equation 17. Typically a lapse rate correction is applied to the measured air temperature in the calculation of deltaT. What's the height of the mean air temperature sensor? More generally, airside transfer is dependent on atmospheric stability. It is typical to convert the airside transfer velocities to neutral transfer velocities. This doesn't affect the authors pair-wise comparisons (e.g. kSO2 vs kSH), but does affect the kSO2 vs u* relationship, for example.

Author response: We revised the calculations as suggested and the text has been

revised as follows: The mean air temperature was corrected for the adiabatic lapse rate, and the sonic temperatures were corrected for humidity. $SO_2$, water vapor, temperature, and winds were corrected to 10m height and neutral stability using COARE (Businger et al., 1971, Fairall et al., 1996, Edson et al., 2013, Fairall et al., 2003).

RC1: P. 9, line 1. How did the authors arrive at a cut off frequency of 1.5 Hz? What is the instrument response time?

Author response: We revised the text to clarify the procedure used to characterize instrument time response as follows: "The attenuation of chemical fluctuations in the inlet were characterized by interrupting the addition of an SO2 gas standard to the air flow, resulting in an exponential decay of the $SO_2$ signal. A decay constant (K) was obtained from the slope of a linear regression to a plot of log(SO2) vs. time. The attenuation of the inlet was modeled as a 1st order low-pass Butterworth filter with a cut-off frequency, Fc=K/(2p), of about 1.5 Hz."

RC1: p. 9, line 16. Have the authors estimated the flux detection limit for their SO2 system? SO2 over the open ocean (especially Southern Hemisphere) is only typically a few tens of ppt.

Author response: The instrument is inherently sensitive enough to make flux measurements over the open ocean, even at 10 ppt levels. However, at sea there would likely be an additional challenge associated with preventing sea-salt accumulation in the inlet which could to lead to loss of SO2 during sampling. We had inadvertently omitted the instrument sensitivity from the manuscript and added the following sentence to the section on $SO_2$ detection: "The $SO_2$ instrument has a sensitivity of approximately 150 Hz ppt-1."

RC1: p. 9, line 18. Reasoning and reference for excluding data with Z/L > 0.07?

Author response: This cutoff is based on an observed inflection in the relationship between TKE and z/L as noted by Oncley et al. (1996). The text was revised as fol-

lows: "4. Stable atmospheric conditions - Intervals with stable atmospheric conditions, defined as z/L > 0.07 were rejected (Oncley et al., 1996)."

RC1: P. 9. Line 19. What were the thresholds used for excluding SO2 ship spikes?

Author response: There was no specific threshold – ship spikes were identified subjectively. The text has been modified to note this.

RC1: p. 10, line 25. There is very large variability in the kmom vs. U10 relationship here. Typically over the open ocean, the relative standard deviation of u* decreases with increasing wind speed. Could some of the variability here be due to tidal height changes or the wind direction being on the edge of the acceptable sea-sector? Also, see my comment about U10 earlier.

Author response: We agree that these factors could contribute to variability in the relationship between U and u*. We prefer not to revise the manuscript as we have no evidence for a specific cause.

RC1: P. 11 line 13. At the beginning of this section, I think the authors should first compare their kSO2 vs. u* relationship to a) the only previous measurements of kSO2 (Faloona et al. J. Atmos Chem 2010), and b) kmethanol from Yang et al. GTWS 2016.

Author response: We added two paragraphs and a figure (Fig. 6) comparing our data with those prior measurements. The following paragraphs were added to the Discussion section: "Faloona et al. (2009) reported airborne eddy covariance measurements of $SO_2$ deposition over the equatorial Pacific. The data from their lowest flight altitude of 30m should be comparable to the data from this study. We made this comparison as a function of u* rather than wind speed to account for the differences in sea surface roughness between the coastal and open ocean environments. The $SO_2$ transfer velocities reported by Faloona et al. (2009) were roughly half those observed at Scripps over a similar range of wind stress (Fig. 6, Table 4). This difference is considerably larger than expected from the scatter in the data or estimated uncertainties in the flux

measurements. Further investigation is needed in order to determine whether a systematic difference exists in $SO_2$ deposition to coastal vs. open ocean waters and, if so, what the cause might be." "A few studies of direct air/sea exchange of highly soluble organic compounds have also been carried out. Fluxes of acetone to the Pacific ocean were reported by Marandino et al. (2005) and methanol fluxes to the Atlantic ocean were reported by Yang et al. (2013). Surprisingly, the direction and/or magnitude of air/sea fluxes observed in those studies were not consistent with observed air/sea concentration differences based on bulk air and seawater measurements. Both studies speculated that this was due to near surface water-side gradients, because assuming a zero sea surface concentration gave reasonable gas transfer velocities with linear wind speed dependence. For acetone, the resulting gas transfer velocities were considerably lower than those observed in this study (Fig. 6, Table 4). For methanol, the gas transfer velocities were similar to this study, but with a slightly stronger dependence on wind stress. The anomalous behavior of acetone and methanol are generally thought to be related to near surface biological or photochemical processes. The presumed near surface gradients are problematic in that they 30 require strong localized production/loss processes and have not yet been observed in the field. Given the uncertainty introduced by these inferred gradients, more detailed analysis of the similarities and differences in the data seem unwarranted."

Figure 6 is shown at the end of this comment. The actual caption for Fig. 6 will read: "Gas transfer velocities as a function of friction velocity for this study and prior measurements of air/sea exchange of highly soluble, air-side controlled gases from Yang et al., 2013, Faloona et al., (2009), Marandino et al. (2005) and this study. The grey line is the COAREG model calculated with the drag coefficents measured during this study, using the Sc number of $SO_2$."

RC1: p. 11, line 18. The authors' assertion that kSO2 is more precise than kSH and kH2O appears to be backed up by their Fig. 4. With increasing wind speed, the scatter in kSO2 only increases marginally (the relative standard deviation probably decreases).

In contrast, the scatter in kSH and kH2O increase substantially with wind speed.

Author response: We agree and added the following sentence to the manuscript: "The transfer velocities for $SO_2$ had significantly less scatter compared to the water vapor and sensible heat transfer velocities at high wind speeds (Figure 4)."

RC1: p. 11, line 25. Technically the name COAREG started with Fairall et al. 2011 JGR (the gas transfer version), not Fairall et al. 2000.

Author response: Fairall et al. (2011) reference added.

RC1: p. 11, line 28-30. It looks like the author substituted computed Cd with the mean Cd v. U10 relationship from the measurements? Would the authors be able to explain more of the variability in the other k data if they prescribe the model with the measured Cd on a point-by-point basis?

Author response: The modeled k's were calculated using the measured Cd on a point by point basis. Just for clarity, we opted to show the linear regression of kmodeled vs u* instead of the individual points. This has been clarified in the figure 4 caption as follows: "Figure 4. Transfer velocities measured at Scripps Pier as a function of wind and friction velocity. Top row: water vapor, sensible heat, and $SO_2$ as a function of $U_{10}$ (black dots). Bottom row: water vapor, sensible heat, and $SO_2$ as a function of $u_*$ with linear least squares regressions and 95% confidence intervals (black dots and black line). Red lines are a second order least squares regression of transfer velocities computed with the COAREG parameterization using measured drag coefficients (Fairall et al., 2000, 2011). Blue lines are transfer velocities computed with COAREG parameterization allowing the model to calculate friction velocities and drag coefficients."

RC1: p. 12, line 16. The authors should specify that these are airside Schmidt numbers, which are largely temperature independent. Also, how does the $Sc_SO2$ here compare with more contemporary predictions (e.g. from Johnson 2000 Ocean Science)?

Author response: The manuscript and caption of Table 3 were revised to specify that these are air-side Schmidt numbers. The differences from Johnson (2000) are negligible. We used the same Fuller et al. (1966) parameterization for diffusivity, which agrees well with measurements. The kinematic viscosity of Hilsenrath (1960) differs by less than 1% from that of Tsilingiris (2008), which was cited by Johnson (2000).

RC1: P. 12 line 19-20. Quoted uncertainties here for the Schmidt number exponents are very large. Are they derived from a regression of rdiffH2O vs. rdiffSO2, etc? There seems to be a lot of variability in the kmom data, which isn't as apparent in the kSO2 data. Is subtracting such a noisy rturb (from kmom) the cause for the poor regression results?

Author response: During revision of the manuscript we took a simpler approach to comparing the gas transfer velocities with each other and with COARE. This eliminated the estimate of Sc number exponent. As a result, the calculations mentioned here is no longer present in the manuscript. The new text is on page 12 and 13.

RC1: p. 12 line 22. More appropriate references than Jaehne et al. 1987 are specific studies of airside transfer, including Hicks et al. 1986 as well as the earlier Liss et al. papers

Author response: During the revisions of the discussion noted above, the text regarding Sc dependence was eliminated.

RC1: p. 23, Fig 4. Show units for slope. Also, typo in caption: 'computed' instead of 'computer'

Author response: Manuscript revised.

RC1: Finally, I personally find 'higher/lower' to be more suitable adjectives for transfer velocity than 'bigger/smaller'. The authors can contact me directly for further questions if they wish.

Author response: We agree. Manuscript revised accordingly.

[Figure]

——end of author response to Reviewer 1——

Author responses to comments by Reviewer 2:

RC2: For gases like SO2, I think the airside resistances are better represented by
$r_{total} = r_{turbulence} + r_{diffusion} + r_{surface}$

Author response: We agree. We modified equation 4 and added the following paragraph to the background section.

"Interfacial surface resistance, i.e. resistance to air/sea gas transfer arising from physical/chemical interactions in a molecular scale layer at the surface is included here for completeness. We are aware of no evidence that such processes are important at clean water surfaces for molecules such as $SO_2$ or $H_2O$ (see Section 2.2.3). The sea surface is often 'contaminated' by the presence of organic compounds and particulates collectively referred to as the sea surface (or marine) microlayer. One could hypothesize that a hydrophobic surface film of sufficient coverage and thickness could introduce resistance to the transfer of small polar molecules such as $SO_2$ or $H_2O$, but such effects have not yet been demonstrated. It is well known that the microlayer can alter the surface tension of the sea surface, dampening the formation of capillary waves, and indirectly altering the turbulent and diffusive resistance to transfer of momentum and gases (Frew et al., 1990; Bock and Frew, 1993; Pereira et al., 2016)."

RC2: It seems to me that you should be able to estimate $r_{surface}$ for $SO_2$ as the difference between $r_{total}$ for $SO_2$ and $r_{total}$ for water vapor, since there is no surface resistance effect for water. You can also compare $r_{total}$ for $SO_2$ with $r_a$ from COAREG, which is a stability-corrected estimate of the turbulent and molecular diffusion resistances but does not consider a surface resistance.

Author response: It is not clear what the first sentence proposes here. Using $r_{total}$ for water vapor as a proxy for $SO_2$ would not account for the difference in diffusive transport. For that, a model is required to partition the resistance. In the second part

of this comment, the reviewer suggests that we infer possible surface resistance from the difference between observed and modeled total resistance. This is conceptually appealing, although it implies a degree of confidence in the gas transfer model that we do not necessarily have. Gas transfer models (like COAREG) have never been tested against field data for air-side controlled gases other than water vapor (setting aside methanol and acetone which have unquantifiable uncertainties associated with near surface water side concentration gradients). A second caveat here is the reviewer's comment that "there is no surface resistance for water". That is true for pure water, but the situation hypothesized by the reviewer is that of an organic surfactant layer. Water and $SO_2$ are both small polar molecules and a surface film capable of retarding $SO_2$ transfer could also impede the transfer of water. We certainly agree that the role of surfactants in gas transfer is interesting and potentially important. We carried out the suggested analysis and added a brief statement about this at the end of the discussion.

RC2: P2 Eq. 1: You could point out that the delta C term defined this way means a downward flux is positive, which is typical in literature dealing with deposition to the surface, but opposite the general convention for gas transfer where the upward flux is positive.

Author response: This comment led us to discover some inconsistencies how we represented the delta C terms in various equations. We revised Equation 1 to:

$$F = K\Big(C_w/\alpha - C_a\Big) \tag{1}$$

This convention of upward flux as positive is now used throughout the paper. Equations 16,17 and 19 were revised accordingly.

RC2: P2 Line 20: define solubility as $Cw_o/Ca_o$ to emphasize this is the equilibrium concentration ratio? Personally, I would avoid the term 'Ostwald coefficient' since this is subject to several slightly different and potentially confusing definitions (see R. Battino, Fluid Phase Equilibria, 15, 231-240, 1984) and just call it the dimensionless (liq/gas) solubility.

Author Response: We added the phrase "at equilibrium" (to avoid introducing new new terminology), and removed mention of "Ostwald coefficient".

RC2: P4 line 11: 'This is sufficiently small. . .' ?

Author Response: The discussion of surface resistance was expanded and a new section was added to the Background. The new text is:

"2.2.3 Surface resistance to $SO_2$ deposition

In order for the molecular interface between water and air to play a significant role in air/sea gas transfer, the surface must introduce a resistance comparable to that across the turbulent and viscous layers above it. The surface can be modeled as a diffusive air-side layer with a thickness (L) equal to the mean free path of $SO_2$ in air, about 120 nm. The resistance across a flat planar surface layer can be estimated as:

$$r_{surf} = L/(\gamma D) = 1.2 \times 10^{-7}/(\gamma \times 1.3 \times 10^{-5} \approx 10^{-2}/\gamma \, sm^{-1}$$

where $\gamma$ and D are the accommodation coefficient and molecular diffusion coefficient of $SO_2$ , respectively (Fuller et al., 1966). The time scales associated with turbulent and diffusive transport can be estimated using the COAREG gas transfer model (Fairall et al., 2000). For a height of 10 m and a wind speed of 10 m s$^{-1}$ under neutral conditions, COAREG yields the following:

$$r_{turb} + r_{diff} = 10^2 \text{ s m}^{-1}$$

An accommodation coefficient of $10^{-4}$ would therefore be required in order for resistance at the surface to be comparable to that of the turbulent and diffusive atmosphere above. Laboratory studies of S uptake into clean water droplets suggest that the mass accommodation coefficient is about 0.1 (Worsnop et al., 1989). At this value, the surface resistance is only about 0.1% of the overall resistance. Thus, surface resistance is not expected to play a significant role in air/sea gas transfer across clean water surfaces. The same is likely true for $H_2O$, which is believed to have an accommodation coefficient near unity, although there is considerable scatter in laboratory experiments

(Morita et al., 2004). As noted earlier, the possibility of additional surface resistance for either $SO_2$ or $H_2O$ due to the presence of natural organic marine microlayers cannot be evaluated due to lack of information about their properties."

RC2: Sec 3.2: Were blank measurements conducted and if so, how? In previous work with this method we have used a coil of HCl-washed copper tube to remove $SO_2$ from the sample stream and determine the background signals at m/z 112 and m/z 114. Admittedly, this is not a perfect blank, because removing one reactant from the air sample perturbs the ion-molecule equilibria in the source, such that the background signal you measure in the absence of $SO_2$ may not be exactly the same as the background when $SO_2$ is present. If the concentration of $SO_2$ (ambient + internal standard) is small compared to the $CO_2$ and ozone concentrations, however, this consideration should be minor.

Author Response: Reviewer one asked a similar question and we repeat the inserted text here. "...where $S_{112}$ and $S_{114}$ are the blank corrected mass spectrometer signals. Blanks involved sampling air through a carbonate-impregnated filter to quantitatively remove ambient $SO_2$. Whatman 41 filters for this purpose were soaked in 1% sodium carbonate solution and dried prior to use.

RC2: P7, eq 9: F is being used for both flux and flow which is confusing. Choose another variable in this equation to represent gas flow. . .

Author Response: Good point. The variable used for flow was changed to lower case "f".

RC2: P7, line 30: The correct references for the COARE model are Fairall et al. 1996, Fairall et al. 2003 and Edson et al. 2013. Fairall et al. 2000 deals specifically with gas transfer.

Author Response: References added.

RC2: P8 line eq 12: You might mention that, Fmom is more commonly called the

Reynolds stress (tau).

Author Response: The text has been revised to indicate this as follows: "Fluxes of momentum (Reynolds stress, ), water vapor, sensible heat and $SO_2$ were calculated for each interval according to"

RC2: Sec 5: See comments above. Also, you could mention that a linear wind speed dependence is expected for very soluble gases and has been demonstrated in other studies (i.e. little or no bubble enhancement to k from breaking waves).

Author Response: We prefer not to state that "a linear wind speed dependence is expected" because there are assumptions inherent in this that would require considerable further explanation. The issue of bubble enhancement and the non-linearity of kw is a subject of contention among some in the gas transfer community and because it is a water side issue, we consider it beyond the scope of this paper.

RC2: In comparisons with the physical model I would just use $k_a$ from COAREG and ignore $k_b$ which should not be important and in any case is the more uncertain parameter.

Author Response: That is correct - we used COAREG to calculate only air side resistances, so $k_b$ plays no role in the calculations. $k_b$ is not mentioned in the paper.

RC2: P11 line 25: The other reference for updates to COAREG is Fairall et al., 2011

Author Response: References added.
* * *
Yang et al. (2013)

COAREG

This Study

Marandino et al. (2005)

Faloona et al. (2009)

k (cm s$^{-1}$)

u$_*$

**Fig. 1.** Comparison of gas transfer velocities for highly soluble gases from this and prior studies.

---

## Author Response (AR1)

Eddy flux measurements of sulfur dioxide deposition to the sea surface

Porter et al.

**RC1 Reviewer comments and author responses (in italics):**

P1, line 18. Please give references and indicate ranges. And note that anthropogenic SO2 emissions are changing globally, with for example decreasing trends in Western Europe and North America.

*References have been added and ranges are indicated.*

P2. Line 1-3. Please give references. Yang et al (PNAS 2013, ACP 2014, GTWS 2016) measured the air-sea transfer of methanol and compared its rate to those of momentum and sensible heat. A very similar method of analysis is used here.

*References added.*

P. 2 Line 6-7. The previous sentence just said that the Faloona et al's measurements were in the MBL. Also, it should be 2010, not 2009.

*"Faloona et al. (2009) reported air/sea eddy covariance surface fluxes for $SO_2$ using a fast-response chemical ionization mass spectrometric technique developed by Bandy et al. (2002). To our knowledge these are the only previous eddy covariance measurement of $SO_2$ surface fluxes over the ocean."*

*There appears to be some confusion about the date of this journal article. The doi links to a pdf that indicates the paper was accepted for publication and copyrighted in 2010, but the article appears in the 2009 issue of the Journal of Atmospheric Chemistry. We are inquiring with the publisher to resolve this issue.*

P. 5. Line 12. What's the tidal range at this site? If significant, it'd alter the measurement height above the mean sea level and so the extent of the flux footprint.

*The tidal range over the course of the experiment was 1.69m. The manuscript was changed to include the following.*

*"The sensing regions of the eddy covariance flux package and the air intake for $SO_2$ detection were located approximately 10 m above the sea surface. The sensor height was corrected for changes in tidal range during the experiment."*

P. 7 line 12. How is the SO2 blank measurement (what I assume the authors meant by 'system blanks') made? Since Vd = -flux/[C], any error in mean [C] due to uncertain blank correction

will propagate to Vd. To test, the authors can plot Vd vs. u* and color-code it by [C]. Is there any pattern?

*The text was revised to clarify the procedure for blank measurements, as follows:*

*"...where $S_{112}$ and $S_{114}$ are blank corrected mass spectrometer signals, $f_{std}$ and $f_{total}$ are the gas flow rates of the isotopic standard and inlet and $X_{tank}$ is the molar mixing ratio of $^{34}SO_2$ in the compressed cylinder. Because the air stream was dried in the inlet tube prior to analysis, $X_{SO2}$ represents the mixing ratio of $SO_2$ in dry air. Blanks were obtained by sampling air through a carbonate-impregnated filter to quantitatively remove ambient $SO_2$. Whatman 41 filters for this purpose were soaked in 1% sodium carbonate solution and dried prior to use."*

*We made the plot of Vd vs. u*, color-coded by [C]. No relationship was observed between [C] and the $SO_2$ transfer velocities.*

P. 7 line 20. What's the typical tilt angle, i.e. the angle between the horizontal and the streamline?

*The average tilt angle was 1.3 degrees. The manuscript was changed to include this information as follows.*

*"...rotating the 3-D winds for each flux interval into the frame of reference of the mean winds and to account for tilt in the sonic anemometer (1.3°)..."*

P. 7, line 30. The authors show later that the mean Cd measured at this location is significantly greater than what the COARE model predicts. If so, U10 computed from the COARE model (assuming open ocean Cd) will be in error. In theory, to get a more accurate U10 it's probably better to use an iterative approach to estimate U10 from the measured Cd. Though the difference might not be very big in this case since the measurement height is already at ~10 m above sea level.

*We agree that this would lead to a more accurate calculation of $U_{10}$. As suggested, the difference is quite small, with an average of only about 0.1% difference from the uncorrected wind speeds. Figures 1,3 and 4 and the text in section 3.3 were updated to reflect the new wind speed calculations.*

p. 7 line 32. Last sentence: 'sensible heat' is left out

*Corrected.*

p. 8. Equation 11. The Licor7500 measures mass concentrations rather than the mixing ratios. Thus a 'Webb' correction for air density fluctuation is required for water vapor flux (probably not a big correction). Has this been applied?

*This Webb correction was applied. The text was revised to explicitly indicate that the correction was done as follows.*

*"Water vapor concentrations measured by the LICOR were corrected to account for air density fluctuations and converted to concentration (mol m$^{-3}$)."*

p. 8 Equation 13. Is T here the sonic temperature or the air temperature? The sonic temperature, approximately equal to virtual temperature, is affected by humidity. Thus one needs to apply a latent heat correction to the sonic heat flux to get sensible heat flux. This probably isn't a big correction, and can be achieved by: a) apply a high frequency (e.g. >=5 Hz, if available) humidity correction to the raw sonic temperature data, or b) use the actual latent heat flux (or bulk latent heat flux) to correct the sonic heat flux

*The T in equation 13 is air temperature after applying a high frequency humidity correction to the raw sonic temperature data. The manuscript was revised as follows to indicate this: "...T is the humidity-corrected air temperature ..."*

P. 8, Equation 17. Typically a lapse rate correction is applied to the measured air temperature in the calculation of deltaT. What's the height of the mean air temperature sensor? More generally, airside transfer is dependent on atmospheric stability. It is typical to convert the airside transfer velocities to neutral transfer velocities. This doesn't affect the authors pair-wise comparisons (e.g. kSO2 vs kSH), but does affect the kSO2 vs u* relationship, for example.

*We revised the calculations as suggested and the text has been revised as follows:*

*The mean air temperature was corrected for the adiabatic lapse rate, and the sonic temperatures were corrected for humidity. SO$_2$, water vapor, temperature, and winds were corrected to 10m height and neutral stability using COARE (Businger et al., 1971, Fairall et al., 1996, Edson et al., 2013, Fairall et al., 2003).*

P. 9, line 1. How did the authors arrive at a cut off frequency of 1.5 Hz? What is the instrument response time?

*We revised the text to clarify the instrument response as follows:*

*"The attenuation of chemical fluctuations in the inlet were characterized by interrupting the addition of an SO$_2$ gas standard to the air flow, resulting in an exponential decay of the SO$_2$ signal. A decay constant (K) was obtained from the slope of a linear regression to a plot of log(SO$_2$) vs. time. The attenuation of the inlet was modeled as a 1st order low-pass Butterworth filter with a cut-off frequency, $F_c=K/(2\pi)$, of about 1.5 Hz."*

p. 9, line 16. Have the authors estimated the flux detection limit for their SO2 system? SO2 over the open ocean (especially Southern Hemisphere) is only typically a few tens of ppt.

*The instrument is inherently sensitive enough to make flux measurements over the open ocean, even at 10 ppt levels.  However, at sea there would likely be an additional challenge associated with preventing sea-salt accumulation in the inlet which could to lead to loss of $SO_2$ during sampling.*

*We had inadvertently omitted the instrument sensitivity from the manuscript.  We added the following sentence to the section on $SO_2$ detection:  "The $SO_2$ instrument has a sensitivity of approximately 150 Hz $ppt^{-1}$."*

p. 9, line 18. Reasoning and reference for excluding data with Z/L > 0.07?

*This cutoff is based on an observed inflection in the relationship between TKE and z/L as noted by Oncley et al. (1996).   The text was revised as follows:*

*4. Stable atmospheric conditions - Intervals with stable atmospheric conditions, defined as z/L > 0.07 were rejected (Oncley et al., 1996).*

P. 9. Line 19. What were the thresholds used for excluding SO2 ship spikes?

*There was no specific threshold – ship spikes were identified subjectively. The text has been modified to note this.*

 p. 10, line 25. There is very large variability in the kmom vs. U10 relationship here. Typically over the open ocean, the relative standard deviation of u* decreases with increasing wind speed. Could some of the variability here be due to tidal height changes or the wind direction being on the edge of the acceptable sea-sector? Also, see my comment about U10 earlier.

*We agree that these factors could contribute to variability in the relationship between U and u\*. We prefer not to revise the manuscript as we have no evidence for a specific cause.*

P. 11 line 13. At the beginning of this section, I think the authors should first compare their kSO2 vs. u* relationship to a) the only previous measurements of kSO2 (Faloona et al. J. Atmos Chem 2010), and b) kmethanol from Yang et al. GTWS 2016.

*The following two paragraphs were added to the discussion section of the document and an accompanying Figure 6 and Table 4.*

*Faloona et al. (2009) reported airborne eddy covariance measurements of $SO_2$ deposition over the equatorial Pacific. The data from their lowest flight altitude of 30m should be comparable to the data from this study. We made this comparison as a function of u_ rather than wind speed to account for the differences in sea surface roughness between the coastal and open ocean environments. The $SO_2$ transfer velocities reported by Faloona et al. (2009) were roughly half those observed at Scripps over a similar range of wind stress (Fig. 6, Table 4). This difference is considerably larger than expected from the scatter in the data or estimated uncertainties in the flux measurements. Further investigation is needed in order to determine whether a systematic*

*difference exists in SO₂ deposition to coastal vs. open ocean waters and, if so, what the cause might be.*

*A few studies of direct air/sea exchange of highly soluble organic compounds have also been carried out. Fluxes of acetone to the Pacific ocean were reported by Marandino et al. (2005) and methanol fluxes to the Atlantic ocean were reported by Yang et al. (2013). Surprisingly, the direction and/or magnitude of air/sea fluxes observed in those studies were not consistent with observed air/sea concentration differences based on bulk air and seawater measurements. Both studies speculated that this was due to near surface water-side gradients, because assuming a zero sea surface concentration gave reasonable gas transfer velocities with linear wind speed dependence. For acetone, the resulting gas transfer velocities were considerably lower than those observed in this study (Fig. 6, Table 4). For methanol, the gas transfer velocities were similar to this study, but with a slightly stronger dependence on wind stress. The anomalous behavior of acetone and methanol are generally thought to be related to near surface biological or photochemical processes. The presumed near surface gradients are problematic in that they 30 require strong localized production/loss processes and have not yet been observed in the field. Given the uncertainty introduced by these inferred gradients, more detailed analysis of the similarities and differences in the data seem unwarranted.*

p. 11, line 18. The authors' assertion that kSO2 is more precise than kSH and kH2O appears to be backed up by their Fig. 4. With increasing wind speed, the scatter in kSO2 only increases marginally (the relative standard deviation probably decreases). In contrast, the scatter in kSH and kH2O increase substantially with wind speed.

*We agree and added the following sentence to the manuscript:*

*"The transfer velocities for SO₂ had significantly less scatter compared to the water vapor and sensible heat transfer velocities at high wind speeds (Figure 4)."*

p. 11, line 25. Technically the name COAREG started with Fairall et al. 2011 JGR (the gas transfer version), not Fairall et al. 2000.

*Fairall et al. (2011) reference added.*

p. 11, line 28-30. It looks like the author substituted computed Cd with the mean Cd v. U10 relationship from the measurements? Would the authors be able to explain more of the variability in the other k data if they prescribe the model with the measured Cd on a point-by-point basis?

*The modeled k's were calculated using the measured Cd on a point by point basis. Just for clarity, we opted to show the linear regression of $k_{modeled}$ vs u\* instead of the individual points. This has been clarified in the figure 4 caption as follows.*

*Figure 4. Transfer velocities measured at Scripps Pier as a function of wind and friction velocity. Top row: water vapor, sensible heat, and SO₂ as a function of $U_{10}$ (black dots). Bottom row: water vapor, sensible heat, and SO₂ as a function of u_ with linear least squares egressions*

*and 95% confidence intervals (black dots and black line). Red lines are a second order least squares regression of transfer velocities computed with the COAREG parameterization using measured drag coefficients (Fairall et al., 2000, 2011). Blue lines are transfer velocities computed with COAREG parameterization allowing the model to calculate friction velocities and drag coefficients.*

p. 12, line 16. The authors should specify that these are airside Schmidt numbers, which are largely temperature independent. Also, how does the Sc_SO2 here compare with more contemporary predictions (e.g. from Johnson 2000 Ocean Science)?

*The manuscript and caption of Table 3 were revised to specify that these are air-side Schmidt numbers.*

*The differences from Johnson (2000) are negligible.  We used the same Fuller et al. (1966) parameterization for diffusivity, which agrees well with measurements.  The kinematic viscosity of Hilsenrath (1960) differs by less than 1% from that of Tsilingiris (2008), which was cited by Johnson (2000).*

P. 12 line 19-20. Quoted uncertainties here for the Schmidt number exponents are very large. Are they derived from a regression of rdiffH2O vs. rdiffSO2, etc? There seems to be a lot of variability in the kmom data, which isn't as apparent in the kSO2 data. Is subtracting such a noisy rturb (from kmom) the cause for the poor regression results?

*During revision of the manuscript we took a simpler approach to comparing the gas transfer velocities with each other and with COARE and eliminated the estimate of Sc number exponent. The calculations mentioned here is no longer present in the manuscript. The new text consists of several paragraphs on page 12 and 13.*

p. 12 line 22. More appropriate references than Jaehne et al. 1987 are specific studies of airside transfer, including Hicks et al. 1986 as well as the earlier Liss et al. papers

*During the revisions mentioned above, the text regarding Sc dependence was eliminated.*

p. 23, Fig 4. Show units for slope. Also, typo in caption: 'computed' instead of 'computer'

*Manuscript revised.*

Finally, I personally find 'higher/lower' to be more suitable adjectives for transfer velocity than 'bigger/smaller'. The authors can contact me directly for further questions if they wish.

*We agree.  Manuscript revised accordingly.*
* * *
**RC2 Reviewer comments and author responses (in italics):**

For gases like SO2, I think the airside resistances are better represented by

r_total = r_turbulence + r_diffusion + r_surface

*We agree. We modified equation 4 and added the following paragraph to the background section.*

*Interfacial surface resistance, i.e. resistance to air/sea gas transfer arising from physical/chemical interactions in a molecular scale layer at the surface is included here for completeness. We are aware of no evidence that such processes are important at clean water surfaces for molecules such as $SO_2$ or $H_2O$ (see Section 2.2.3). The sea surface is often 'contaminated' by the presence of organic compounds and particulates collectively referred to as the sea surface (or marine) microlayer. One could hypothesize that a hydrophobic surface film of sufficient coverage and thickness could introduce resistance to the transfer of small polar molecules such as $SO_2$ or $H_2O$, but such effects have not yet been demonstrated. It is well known that the microlayer can alter the surface tension of the sea surface, dampening the formation of capillary waves, and indirectly altering the turbulent and diffusive resistance to transfer of momentum and gases (Frew et al., 1990; Bock and Frew, 1993; Pereira et al., 2016).*

It seems to me that you should be able to estimate r_surface for SO2 as the difference between r_total for SO2 and r_total for water vapor, since there is no surface resistance effect for water. You can also compare r_total for SO2 with r_a from COAREG, which is a stability-corrected estimate of the turbulent and molecular diffusion resistances but does not consider a surface resistance.

*It is not clear what the first sentence proposes here. Using r_total for water vapor as a proxy for $SO_2$ would not account for the difference in diffusive transport. For that, a model is required.*

*In the second part of this comment, the reviewer suggests that we infer possible surface resistance from the difference between observed and modeled total resistance. This is conceptually appealing, although it implies a degree of confidence in the gas transfer model that we do not necessarily share. Gas transfer models (like COAREG) have never been tested against field data for air-side controlled gases other than water vapor (setting aside methanol and acetone which have unquantifiable uncertainties associated with near surface water side concentration gradients). A second caveat here is that the reviewer asserts that "there is no surface resistance for water". That is true for pure water, but the situation hypothesized by the reviewer is that of an organic surfactant layer. Water vapor and $SO_2$ are both small polar molecules and a surface film capable of retarding $SO_2$ transfer could also impede the transfer of water.*

*We do agree that the role of surfactants in gas transfer is interesting and potentially important. We carried out the suggested analysis and added a brief statement at the end of the discussion.*

Other more minor suggestions and comments:

P2 Eq. 1: You could point out that the delta C term defined this way means a downward flux is positive, which is typical in literature dealing with deposition to the surface, but opposite the general convention for gas transfer where the upward flux is positive.

*This comment led us to discover some inconsistencies how we represented the delta C terms in various equations. We revised Equation 1 to:*

$$F = K\left(\frac{C_w}{\alpha} - C_a\right)$$

*This convention of upward flux as positive is now used throughout the paper. Equations 16,17 and 19 were revised to reflect this.*

P2 Line 20: define solubility as Cw_o/Ca_o to emphasize this is the equilibrium concentration ratio? Personally, I would avoid the term 'Ostwald coefficient' since this is subject to several slightly different and potentially confusing definitions (see R. Battino, Fluid Phase Equilibria, 15, 231-240, 1984) and just call it the dimensionless (liq/gas) solubility.

*We removed the mention of 'Ostwald coefficient".*

P4 line 11: 'This is sufficiently small. . .' ?

*The discussion of surface resistance was expanded and a new section was added to the Background. The new text is:*

*2.2.3 Surface resistance to SO₂ deposition*

*In order for the molecular interface between water and air to play a significant role in air/sea gas transfer, the surface must introduce a resistance comparable to that across the turbulent and viscous layers above it. The surface can be modeled as a diffusive air-side layer with a thickness (L) equal to the mean free path of SO₂ in air, about 120 nm. The resistance across a flat planar surface layer can be estimated as:*

$$r_{surf} = \frac{L}{\gamma D} = \frac{1.2 \times 10^{-7}}{\gamma \times 1.3 \times 10^{-5}} \approx \frac{10^{-2}}{\gamma} ms^{-1}$$

*where γ and D are the accommodation coefficient and molecular diffusion coefficient of SO₂ , respectively (Fuller et al., 1966). The time scales associated with turbulent and diffusive transport can be estimated using the COAREG gas transfer model (Fairall et al., 2000). For a height of 10 m and a wind speed of 10 m s⁻¹ under neutral conditions, COAREG yields the following:*

$$r_{turb} + r_{diff} = 10^2 m^{-1}s$$

*An accommodation coefficient of 10⁻⁴ would therefore be required in order for resistance at the surface to be comparable to that of the turbulent and diffusive atmosphere above. Laboratory*

*studies of S uptake into clean water droplets suggest that the mass accommodation coefficient is about 0.1 (Worsnop et al., 1989). At this value, the surface resistance is only about 0.1% of the overall resistance. Thus, surface resistance is not expected to play a significant role in air/sea gas transfer across clean water surfaces. The same is likely true for $H_2O$, which is believed to have an accommodation coefficient near unity, although there is considerable scatter in laboratory experiments (Morita et al., 2004). As noted earlier, the possibility of additional surface resistance for either $SO_2$ or $H_2O$ due to the presence of natural organic marine microlayers cannot be evaluated due to lack of information about their properties.*

Sec 3.2: Were blank measurements conducted and if so, how? In previous work with this method we have used a coil of HCl-washed copper tube to remove SO2 from the sample stream and determine the background signals at m/z 112 and m/z 114. Admittedly, this is not a perfect blank, because removing one reactant from the air sample perturbs the ion-molecule equilibria in the source, such that the background signal you measure in the absence of SO2 may not be exactly the same as the background when SO2 is present. If the concentration of SO2 (ambient + internal standard) is small compared to the CO2 and ozone concentrations, however, this consideration should be minor.

*Reviewer one asked a similar question and we repeat the inserted text here.*

*"...where $S_{112}$ and $S_{114}$ are the blank corrected mass spectrometer signals. Blanks involved sampling air through a carbonate-impregnated filter to quantitatively remove ambient $SO_2$. Whatman 41 filters for this purpose were soaked in 1% sodium carbonate solution and dried prior to use.*

P7, eq 9: F is being used for both flux and flow which is confusing. Choose another variable in this equation to represent gas flow. . .

*Good point. The variable used for flow was changed to lower case "f".*

P7, line 30: The correct references for the COARE model are Fairall et al. 1996, Fairall et al. 2003 and Edson et al. 2013. Fairall et al. 2000 deals specifically with gas transfer.

*References added.*

P8 line eq 12: You might mention that, Fmom is more commonly called the Reynolds stress (tau).

*The text has been revised to indicate this as follows:*

*"Fluxes of momentum (Reynolds stress, $\tau$), water vapor, sensible heat and SO2 were calculated for each interval according to"*

Sec 5: See comments above. Also, you could mention that a linear wind speed dependence is expected for very soluble gases and has been demonstrated in other studies (i.e. little or no bubble enhancement to k from breaking waves).

*We prefer not to include the statement "a linear wind speed dependence is expected" because there are assumptions inherent in this that would require considerable further explanation. The issue of bubble enhancement and the non-linearity of kw is a subject of contention among some in the gas transfer community and is beyond the scope of this paper.*

In comparisons with the physical model I would just use k_a from COAREG and ignore k_b which should not be important and in any case is the more uncertain parameter.

*As noted by the reviewer, we used COAREG to calculate only air side resistances, so k_b plays no role in the calculations.*

P11 line 25: The other reference for updates to COAREG is Fairall et al., 2011

*Reference added.*

[revised manuscript text omitted]